# Analysis of Spatiotemporal Variation in Habitat Suitability for *Oedaleus decorus asiaticus* Bei-Bienko on the Mongolian Plateau Using Maxent and Multi-Source Remote Sensing Data

**DOI:** 10.3390/insects14060492

**Published:** 2023-05-24

**Authors:** Fu Wen, Longhui Lu, Chaojia Nie, Zhongxiang Sun, Ronghao Liu, Wenjiang Huang, Huichun Ye

**Affiliations:** 1College of Water Resources Science and Engineering, Taiyuan University of Technology, Taiyuan 030024, China; 2International Research Center of Big Data for Sustainable Development Goals, Beijing 100094, China; 3Key Laboratory of Digital Earth Science, Aerospace Information Research Institute, Chinese Academy of Sciences, Beijing 100094, China; 4Key Laboratory of Earth Observation of Hainan Province, Hainan Aerospace Information Research Institute, Sanya 572029, China; 5China Agricultural Museum, Beijing 100125, China

**Keywords:** Mongolian Plateau, *O. decorus asiaticus*, Maxent model, grasshopper, habitat suitability, spatiotemporal characteristics

## Abstract

**Simple Summary:**

*Oedaleus decorus asiaticus* Bei-Bienko is one of the most dangerous agricultural pests on the Mongolian Plateau. Due to its small size, it is difficult to monitor and control. Therefore, it is important to detect the key factors affecting the spatial distribution of grasshopper occurrence, to extract the suitable areas for grasshoppers, and to analyze the spatial and temporal evolution of the suitable areas for grasshoppers for monitoring and early warning of grasshoppers on the Mongolian plateau. Here the spatiotemporal variation in the habitat suitability for *O. decorus asiaticus* on the Mongolian Plateau was assessed using maximum entropy (Maxent) modeling along with multi-source remote sensing data and geospatial data.

**Abstract:**

*O. decorus asiaticus* is a major grasshopper species that harms the development of agriculture on the Mongolian Plateau. Therefore, it is important to enhance the monitoring of *O. decorus asiaticus*. In this study, the spatiotemporal variation in the habitat suitability for *O. decorus asiaticus* on the Mongolian Plateau was assessed using maximum entropy (Maxent) modeling along with multi-source remote sensing data (meteorology, vegetation, soil, and topography). The predictions of the Maxent model were accurate (AUC = 0.910). The key environmental variables affecting the distribution of grasshoppers and their contribution were grass type (51.3%), accumulated precipitation (24.9%), altitude (13.0%), vegetation coverage (6.6%), and land surface temperature (4.2%). Based on the assessment results of suitability by Maxent model, the model threshold settings, and the formula for calculating the inhabitability index, the 2000s, 2010s, and 2020s inhabitable areas were calculated. The results show that the distribution of suitable habitat for *O. decorus asiaticus* in 2000 was similar to that in 2010. From 2010 to 2020, the suitability of the habitat for *O. decorus asiaticus* in the central region of the Mongolian Plateau changed from moderate to high. The main factor contributing to this change was accumulated precipitation. Few changes in the areas of the habitat with low suitability were observed across the study period. The results of this study enhance our understanding of the vulnerability of different regions on the Mongolian Plateau to plagues of *O. decorus asiaticus* and will aid the monitoring of grasshopper plagues in this region.

## 1. Introduction

Grasshoppers can have major deleterious effects on the agriculture and animal husbandry industries because of their explosive growth, destructive capabilities, and migratory behavior [1]. Meanwhile, outbreaks of grasshoppers can induce major harm to the environment and human society, such as grassland degradation and desertification [2,3,4,5].

The Mongolian Plateau is one of the largest plateaus in the world. Grassland is the dominant ecosystem on the Mongolian Plateau, covering approximately 1.1 million km^2^. The Mongolian Plateau not only is an important ecological barrier in East Asia, but it also plays an important role in the global carbon cycle [6]. It is one of the three most important regions for animal husbandry in the world, the annual production of various types of livestock in this region amounts to approximately 120 million head, animal husbandry is the main sector of the economy on the Mongolian Plateau, and it maintains the livelihoods of approximately 10 million low-income people [7]. However, management is frequently not optimal in this vast area of grassland, and grasshopper populations can be sufficiently dense to result in additional habitat degradation [8]. Therefore, it is necessary to use multi-source remote sensing data to infer the distribution of grasshoppers and predict the grasshopper outbreak areas in order to formulate countermeasures in advance to prevent the occurrence of grasshoppers and reduce the damage of grasshoppers to the local ecological environment and social economy.

*O. decorus asiaticus* belongs to the family Oedipodidae, order Orthoptera; it is one of the most dominant grasshopper species on the Mongolian Plateau [8,9]. This species induces major damage to *Leymus chinensis*, *Stipa*, and other plants [10]. Outbreaks of *O. decorus asiaticus* can lead to rapid reductions in the areas of pasture and grassland degradation [11]. *O. decorus asiaticus* has become an indicator species of grassland degradation [12]. Controlling *O. decorus asiaticus* becomes extremely difficult once swarms begin to engage in collective behavior because of their long-distance dispersal capacities, destructive capabilities, and explosive population growth [13]. Characterizing the habitat suitability for this grasshopper species on the Mongolian Plateau is thus critically important for developing methods to minimize the impacts of future outbreaks.

The distributions of grasshoppers cannot be characterized using satellite remote sensing images because of their small size [14]. Recent studies have shown that the growth and development of grasshoppers are closely related to their habitat [15], including meteorological, topographical, soil, and vegetation factors [16,17,18,19,20]. These factors affect the distribution, outbreak probability, and migration behavior of grasshoppers [21,22]. Temperature and precipitation are the main meteorological factors affecting grasshoppers [23,24,25]. For example, high temperatures accelerate the hatching of grasshopper eggs, which affects the reproductive output of grasshoppers [26]. Temperature changes can also affect grasshopper survival [27]. Precipitation mainly affects the growth and development of grasshoppers through its effects on soil moisture [28,29]. Altitude and slope are the main topographic factors affecting the distribution of grasshoppers [30,31,32,33,34]. The soil type and soil water content also affect the distribution and densities of grasshoppers [28,35]. In addition, the local composition of plants and vegetation coverage affect the growth and development of grasshoppers [36,37,38]. The above studies have shown that the occurrence of grasshoppers is determined by multiple habitat factors [22]. Therefore, comprehensive analyses of the effects of various habitat factors on the occurrence of grasshoppers are needed [39,40]. Preliminary studies of the effects of changes in habitat on the occurrence of grasshoppers are needed to evaluate the vulnerability of different regions to damage from grasshoppers.

Traditionally, the occurrence of grasshoppers is evaluated by collecting data manually, which is time consuming and laborious. The wide distribution of grasshoppers, especially in remote areas, poses a major challenge to field surveys. There is thus a need to develop approaches that permit the collection of data on grasshopper habitats to evaluate the suitability of the habitat in different areas for grasshoppers. The development of remote sensing technology has increased the availability of various types of data at high spatial and temporal resolutions [41]. Remote sensing technology can be used to accurately monitor the occurrence of grasshoppers by permitting changes in habitat conditions to be characterized [42]. Consequently, remote sensing data have been increasingly used to monitor changes in habitat suitability for grasshoppers [43,44,45].

Niche models have been widely used to study the distribution of suitable habitats for plants and animals, as well as monitor pest populations and species invasions [46]. The development of geographic information technology and niche models has increased the use of genetic algorithm for rule set production models, ecological niche factor analysis models [47], and maximum entropy (Maxent) models [48] for evaluating the habitat suitability for various species. Maxent models are the most widely used niche models. Maxent models are based on Maximum entropy theory, which involves predicting the potential distribution of species using species occurrence records and environmental variables [49]. The model can also simplify complex natural systems, which allows robust results to be obtained with small sample sizes [50,51]. Therefore, Maxent models, along with key habitat factors and remote sensing data, could be used to evaluate and predict the distribution of suitable habitat for grasshoppers.

Here the integration of the Maxent model and remote sensing data is used to conduct a species–environmental matching model. This study aims to (1) determine the distribution of the *O. decorus asiaticus* inhabitable area on the Mongolian plateau; (2) identify key environmental factors affecting the distribution of *O. decorus asiaticus* on the Mongolian plateau; and (3) analyze the spatial and temporal evolution of the *O. decorus asiaticus* inhabitable area on the Mongolian plateau in 2000, 2010, and 2020 and identify the main driving factors leading to changes in the *O. decorus asiaticus* inhabitable area on the Mongolian plateau. The results of this study enhance our understanding of grasshopper distribution on the Mongolian plateau and will provide guidance for the monitoring and management of grasshoppers on the Mongolian plateau.

## 2. Materials and Methods

### 2.1. Study Area

The Mongolian Plateau is one of the largest plateaus in the world. The terrain is high in the west and low in the east. The Mongolian Plateau includes Mongolia, Inner Mongolia, and southern Siberia in Russia. Our study area included Inner Mongolia and Mongolia (Figure 1); the main areas with grasshoppers on the Mongolian Plateau were between 37° and 54° N and between 87° and 126° E. The study area experienced a temperate continental climate. The average winter temperature was −26 °C, and the average summer temperature was 17 °C. The average annual precipitation was approximately 200 mm. In the study area, the habitat transitioned from meadow steppe to typical steppe and desertified steppe from northeast to southwest. The soil types in the study area were mainly chernozem, chestnut soil, brown calcium soil, and gray–brown desert soil [6,7]. Our study was focused on *O. decorus asiaticus*. The locations of *O. decorus asiaticus* outbreaks are shown in Figure 1. 

### 2.2. Data Acquisition and Processing

#### 2.2.1. Grasshopper Survey Data

A regional survey method according to the standard in agricultural industry of the People’s Republic of China (NY/T 1578-2007, Rules for investing locality and grasshopper in grassland) was used to investigate the overall grasshopper occurrence. In brief, a total of 1745 grasshopper occurrence points were obtained by the route survey method for spring, summer, and autumn seasons from 2018 to 2022, and the distance between sampling points was not less than 100 m. The survey routes covered all major geomorphic units and areas vulnerable to grasshopper outbreaks. The number and location of grasshoppers were recorded during the surveys, and absence data were not recorded. The grasshopper occurrence points were mainly distributed in the southeastern part of the Mongolian Plateau (Figure 1), where grasshopper infestations were most frequent. The latitude and longitude information of the grasshopper points were saved in .csv format for subsequent use in Maxent modeling.

#### 2.2.2. Habitat Indicators Based on Remote Sensing and Other Geospatial Data

The growth and development of grasshoppers were affected by the weather, topography, soil, and vegetation. The effects of these habitat factors on *O. decorus asiaticus* varied among growth periods. The life cycle of *O. decorus asiaticus* included the spawning period (July–September), the overwintering period (November–March), the hatching period (April–May), and the growing period (June). Among them, the nymph and emergence periods are in May to August. The altitude, slope, grass type, soil type, vegetation coverage, aboveground biomass, accumulated precipitation, and land surface temperature affected the spawning site selection of grasshoppers [52,53,54]. During the overwintering period, extremely low temperatures reduced the survival of grasshopper eggs [44]. In the hatching period, aside from low temperatures, which could induce the death or diapause of grasshopper eggs [55], the soil type and precipitation could affect the hatching success rate of grasshopper eggs [14]. In the nymph and emergence periods, the food and growth environment determined whether grasshoppers could grow normally [14].

To test for strong collinearity between variables, we performed a Pearson correlation analysis for each variable (Table 1). The statistical analysis was implemented in SPSS Statistics 22.0 (IBM Corp, Armonk, NY, USA). The results of the analysis showed that vegetation coverage was strongly correlated with aboveground biomass and land surface temperature. However, the purpose of our simulation by Maxent was to evaluate the weights of each habitat factor of grasshopper. Therefore, we used the annual means of the following eight environmental variables to evaluate the area of suitable habitat for *O. decorus asiaticus*: altitude, slope, grass type, soil type, vegetation coverage, aboveground biomass, accumulated precipitation, and land surface temperature (Table 2). This includes the main habitat factors affecting the growth and survival of *O. decorus asiaticus* throughout its entire life cycle.

Combining the environmental factors affecting the growth and development of grasshoppers on the Mongolian Plateau, the remote sensing data used in this study were mean vegetation coverage from May to August, mean aboveground biomass from May to August, mean annual land surface temperature from January to December, and mean annual accumulated precipitation from January to December. The spatial resolution of vegetation coverage and aboveground biomass was 1 km, and the temporal resolution was 16 days; they were calculated using the normalized difference vegetation index (*NDVI*) data in the MOD13 A2 product. The formula was as follows:(1)$$VC=\left(NDVI-NDVI_{min}\right)/\left(NDVI_{max}-NDVI_{min}\right)$$
where *VC* is the vegetation coverage, which takes a value from 0 to 1; *NDVI* is the normalized vegetation index; $NDVI_{max}$ is the maximum *NDVI*; and $NDVI_{min}$ is the minimum *NDVI*.
(2)AGB=277.7×NDVI

In the formula, *AGB* is the aboveground biomass, and *NDVI* is the normalized vegetation index.

The spatial resolution of the land surface temperature data was 1 km, and the temporal resolution of these data was 8 days; the land surface temperature data were calculated using the LST_Day_1 km and LST_Night_1 km data in MOD11 A2 products. The accumulated precipitation data were calculated from the total _ precipitation data in the ERA5-Land product, which were hourly data with a spatial resolution of 11 km.

Other geospatial data used in this study included the altitude, slope, soil type, and grass type data. The altitude was derived from NASA DEM data (https://e4ftl01.cr.usgs.gov/MEASURES/NASADEM_HGT.001/, accessed on 20 November 2022) with a spatial resolution of 1 km. The soil type data were downloaded from GEE with a spatial resolution of 250 m from the United States Department of Agriculture (https://www.usda.gov, accessed on 20 November 2022). The grass-type data were collected from the National Tibetan Plateau Data Center (http://data.tpdc.ac.cn, accessed on 20 November 2022), which are secondary data based on the European Space Agency Globe Cover 2009 dataset with a spatial resolution of 1 km. The slope data were calculated using ArcGIS based on the DEM data, and the spatial resolution of these data was 1 km. To distinguish the steppe areas of the Mongolian plateau from the non-steppe areas, the Globe Cover 2009 dataset with 0 and 1 attributes was used to mask all the data (0 for non-grassland areas, 1 for grassland areas).

#### 2.2.3. Maxent Model

Maxent (version 3.4.1) downloaded from the American Museum of Natural History (https://biodiversityinformatics.amnh.org/open_source/maxent/, accessed on 10 December 2022) was used to analyze the area of suitable habitat for *O. decorus asiaticus* on the Mongolian Plateau. The Maxent model is based on maximum entropy theory, which evaluates the distribution of suitable habitat according to species occurrence records and environmental factors [49,56]. The Maxent model shows high performance in modeling species distributions and analyzing the relationship between species and environmental variables [57,58]. The Maxent model formula is as follows [49]:(3)$$P_{w}(y|x)=\frac{1}{Z_{w}\left(x\right)}\exp\left(\overset{n}{\underset{i=1}{\sum }}w_{i}f_{i}\left(x,y\right)\right)$$
(4)$$Z_{w}\left(x\right)=\underset{y}{\sum }\exp\left(\overset{n}{\underset{i=1}{\sum }}w_{i}f_{i}\left(x,y\right)\right)$$
where x is the input environment variable, y is the geographical location of the grasshopper occurrence area, $f_{i}\left(x,y\right)$ is the characteristic function, $w_{i}$ is the weight of the characteristic function, n represents the number of datasets, and $P_{w}(y|x)$ represents habitat suitability.

The coordinate system, data format, grid size, row and column number, processing range, and spatial resolution (1 km) were made the same for the vegetation coverage, aboveground biomass, land surface temperature, accumulated precipitation, altitude, slope, grass type, and soil type data. Finally, the data in grid form were converted into ASCII files, which are required for Maxent modeling. All these processes were unified through the built-in Python plug-in of ArcGIS. The grasshopper occurrence and environmental variable data were imported into the Maxent model, and the soil type and grass type were discrete. The used model training methods included the linear, quadratic, hinge, product, and auto programs [59]; our model compared several combinations of regularization parameters and selected the best ones. In the environmental parameter settings, the importance of the environmental variables was evaluated using the jackknifing method, and the procedure was repeated 15 times. During each run, 70% of the occurrence records were used as the training set, and 30% of the occurrence records were used as the test set; these points were selected randomly and uniformly by the model, which could ensure spatial independence to some extent. The default values were used for the remaining parameters.

The accuracy of the Maxent model was evaluated using the area under the curve (AUC) of the receiver operating characteristic (ROC) curve, omission curves, and TSS. The value of the AUC ranged from 0 to 1. The closer the values of AUC and TSS were to 1, the higher the accuracy of the model. The closer the test omission in the omission curve was to the predicted omission, the better the indicated simulation result. In addition, the variable contribution percentage was used to identify the key environmental factors affecting the habitat suitability; the response curve was used to clarify the relationships between environmental factors and habitat suitability probability.

#### 2.2.4. Calculation of Habitat Suitability for Grasshoppers

The weights and thresholds of key habitat factors contributing to habitat suitability for grasshoppers were simulated using 2017–2021 data. The above weights and thresholds of the habitat factors were used to calculate the habitat suitability in 2000, 2010, and 2020. The formula for calculating the inhabitability index (*IH*) was as follows [8]:(5)$$IH\left(x,y\right)=\overset{n}{\underset{i=1}{\sum }}E_{i}\left(x,y\right)W_{i}$$
where $IH\left(x,y\right)$ is the suitable index at $\left(x,y\right)$, n is the number of habitat factors (five in this study), $E_{i}\left(x,y\right)$ is the ith habitat factor value (i=1, 2, 3, ……n) at point $\left(x,y\right)$, and $W_{i}$ is the weight of the ith habitat factor.

In this study, the suitable threshold of the Maxent model was set to 0.5. Based on the literature and expert experience [8,14,60,61], the suitability of each habitat factor was divided into low suitability (<0.5), medium suitability (0.5–0.6), and high suitability (>0.6). In calculations, areas with low suitability were assigned values of 1, areas with medium suitability were assigned values of 2, and areas with high suitability were assigned values of 3. The grasshopper inhabitability index (*IH*) was obtained via weighted summation, and values of the *IH* ranged from 1 to 3 (low to high). Based on the literature and expert experience [14,60], we divided the suitable habitat for grasshoppers into three levels: level 1 (*IH* ≥ 2.5) indicated high suitability, which was an area that was highly vulnerable to experiencing major damage from grasshoppers; level 2 (1.5 ≤ *IH* < 2.5) indicated medium suitability, which was an area vulnerable to experiencing damage from grasshoppers; and level 3 (1.0 ≤ *IH* < 1.5) indicated low suitability, which was an area that was not vulnerable to experiencing damage from grasshoppers. A summary of the scheme used to classify areas in terms of habitat suitability and risk from grasshopper damage is shown in Table 3.

## 3. Results

### 3.1. Assessment of the Habitat Suitability for Grasshoppers on the Mongolian Plateau

#### 3.1.1. Accuracy of Maxent Models

A total of 15 simulations were carried out in this study. The simulation results showed that the maximum training AUC was 0.916, the minimum was 0.912, and the average training AUC was 0.914; the maximum test AUC was 0.917, the minimum was 0.903, the average test AUC for the replicate runs was 0.910, and the standard deviation was 0.004. In addition, the average TSS value for 15 randomly repeated runs was 0.631, and the mean omission on test samples was close to the predicted omission (the black straight line). These results indicated that the simulation results were fair (Figure 2).

#### 3.1.2. Percentage Contribution of Each Habitat Factor to Determining Habitat Suitability

The weight of each habitat factor was determined using the variable contribution percentage (Table 4). The weights of the habitat factors were as follows: grass type (49%), accumulated precipitation (23.8%), altitude (12.4%), vegetation coverage (6.3%), land surface temperature (4%), soil type (3.2%), slope (1.2%), and aboveground biomass (0.1%). The cumulative percentage contribution of grass type, accumulated precipitation, altitude, vegetation coverage, and land surface temperature were 95.5%. The importance of each habitat factor in Table 3 indicated that the soil type, slope, and aboveground biomass had little effect on the habitat suitability for grasshoppers. Therefore, the grass type, accumulated precipitation, altitude, vegetation coverage, and land surface temperature were sufficient for evaluating the habitat suitability for grasshoppers on the Mongolian Plateau. A follow-up analysis was conducted using these five key environmental indicators. The original weights of these five key habitat factors were multiplied by 100/95.5 to obtain the final weights (Table 5). The weight of each factor was as follows: grass type (51.3%), accumulated precipitation (24.9%), altitude (13.0%), vegetation coverage (6.6%), and land surface temperature (4.2%).

#### 3.1.3. Classification of Key Habitat Factors Determining Habitat Suitability

Figure 3 shows the response curve, which indicates the relationships between grasshopper suitability and key habitat factors on the Mongolian Plateau. The classification threshold of each habitat factor was determined based on the default setting of the Maxent model (suitability threshold of 0.5), the literature, and expert experience [14,62]. The suitability of each habitat factor was divided into three levels: low (<0.5), medium (0.5–0.6), and high (>0.6). According to the classification results of the habitat factors (Table 6), the habitat suitability was highest when the grass types were temperate desertified grassland, temperate steppe desert, and temperate desert; the altitude was between 1131 m and 1404 m; the vegetation coverage was between 0 and 0.59; the accumulated precipitation was between 404 mm and 489 mm; and the land surface temperature was between 5.38 °C and 8.17 °C.

### 3.2. Habitat Suitability for Grasshoppers on the Mongolian Plateau

Based on the assessment results of suitability by the Maxent model, the model threshold settings, and the formula for calculating the inhabitability index, the 2000s, 2010s, and 2020s inhabitable areas were calculated (Figure 4). The results showed that the areas with suitable habitat for grasshoppers on the Mongolian Plateau were similar in 2000 and 2010, areas with highly suitable habitat were mainly located in the southeastern portion of the Mongolian Plateau, and a small number of highly suitable areas were present in the northern and northwestern portions of the Mongolian Plateau. Areas with moderately suitable habitat were mainly distributed in the eastern and northern regions of the Mongolian Plateau. Areas with low suitability were mainly present in the northeastern and northwestern regions of the Mongolian Plateau. In 2020, areas with highly suitable habitat were mainly distributed in the eastern part of the Mongolian Plateau, and highly suitable areas were also present in the southeastern and northern regions. Moderately suitable areas were mainly distributed in the northern and southeastern regions. Areas with low suitability were mainly distributed in the northeastern and northwestern portions of the Mongolian Plateau.

The grassland area of the Mongolian Plateau is approximately 1.102 million km^2^, which accounts for approximately 40.44% of the total area of the study area. In 2000, the areas with high, medium, and low suitability accounted for 16.69%, 63.59%, and 19.72% of the total grassland area, respectively. In 2010, the areas with high, medium, and low suitability accounted for 13.87%, 66.41%, and 19.72% of the total grassland area, respectively. In 2020, the areas with high, moderate, and low suitability accounted for 27.31%, 51.57%, and 21.12% of the total grassland area, respectively.

### 3.3. Spatial and Temporal Variability in Habitat Suitability for Grasshoppers on the Mongolian Plateau and the Effects of Environmental Factors

Upon visual inspection, the distribution of suitable habitat for grasshoppers on the Mongolia Plateau was similar in 2000 and 2010, but the distribution of suitable habitat for grasshoppers in 2020 was qualitatively different in the mapped suitability levels from that in 2000 and 2010 (Figure 4). A large number of moderately suitable areas in the eastern and northern portions of the Mongolian Plateau in 2000 and 2010 became highly suitable areas in 2020.

Areas with high, medium, and low suitability were 2.82% lower, 2.82% higher, and the same in 2010 compared with 2000. Areas with high, medium, and low suitability were 13.44% higher, 14.84% lower, and 1.4% higher in 2020 compared with 2010. Overall, there was little change in the total area of habitat with low suitability across the three periods. Changes in the area of moderately suitable habitat to highly suitable habitat from 2010 to 2020 were more pronounced than such changes from 2000 to 2010.

According to the classification of habitat factors in 2000, 2010, and 2020 (Figure 5), there were qualitatively similar average vegetation coverages of the same grass types and at the same altitudes. The differences in the distribution of habitat suitability for grasshoppers in the three periods were mainly affected by the land surface temperature and accumulated precipitation. Changes in the land surface temperature between 2000 and 2010 led to a decline in grasshoppers’ suitability in parts of the northern Mongolian Plateau. In contrast, changes in the land surface temperature between 2010 and 2020 resulted in increased grasshoppers’ suitability in parts of the northern Mongolian Plateau but reduced grasshoppers’ suitability in parts of the central Mongolian Plateau. The suitability classification of the accumulated precipitation was similar in 2000 and 2010. The shift in the suitability of the accumulated precipitation for grasshoppers from low and medium suitability to high suitability was more pronounced from 2010 to 2020 compared with that from 2000 to 2010 in the eastern Mongolian Plateau.

## 4. Discussion

The Mongolian Plateau is a large area with a high variation in climate: meteorological factors such as precipitation and temperature vary extensively. The variation in soil, vegetation, and terrain is also complex. This results in variation in the habitat suitability for grasshoppers in different regions of the Mongolian Plateau. Acquiring field survey data of grasshoppers in the study region is difficult because of the complexity of the terrain and migratory characteristics of grasshoppers. These pose major challenges for studies of habitat suitability for grasshoppers on the Mongolian Plateau. The development of remote sensing technology has increased the availability of data with high spatial and temporal resolution [41], and this has facilitated studies aimed at predicting habitat suitability for grasshoppers in this region.

Maxent models are based on Maxent theory, and they can be used to predict the potential distribution of species through occurrence records and environmental variables [49]. Maxent models simplify complex natural systems and permit reliable predictions to be obtained with small sample sizes [50,51]. The results of the Maxent model simulation showed that the average AUC value for 15 repeated runs was 0.910, and the TSS value was 0.631. Meanwhile, the average omission rate of the test sample was close to the predicted omission rate as shown by the results of the omission curve. The above results indicated that the model had a good performance. However, since grasshoppers are an outbreak species, the Maxent model assumed that species were in equilibrium; this may reduce the accuracy of the model. In addition, since our model was for *O. decorus asiaticus* and was limited to the spatial extent of the Mongolian plateau, there was a real limitation in model extrapolation. If one wants to extrapolate the model, one needs to adjust the extrapolation area; for example, Multi-surface Environmental Similarity Surfaces (MESS) could be computed [63].

The cumulative percentage contribution of grass type, accumulated precipitation, altitude, and vegetation coverage was 91.5%. The percentage contribution of the land surface temperature was only 4%, but its importance was high. This might stem from the temperate continental climate of the Mongolian Plateau. There is a large temperature difference between winter and summer; extreme cold weather can occur in winter, and low temperatures hinder the growth and development of grasshoppers and the hatching of grasshopper eggs [26]. Some studies have shown that the soil moisture and temperature have a greater effect on grasshoppers during the egg-laying, hatching, and cyst stages [64]. Therefore, the grass type, accumulated precipitation, altitude, vegetation coverage, and land surface temperature were the key habitat factors determining the suitability of habitat for grasshoppers on the Mongolian Plateau. The effects of the soil type, slope, and aboveground biomass on the habitat suitability for grasshoppers on the Mongolian Plateau were weaker than the five key habitat factors.

In this study, no significant difference in the distribution of areas with suitable habitats for grasshoppers was observed between 2000 and 2010. Only slight changes were observed in medium and highly suitable areas, and areas with low suitability remained unchanged. The low variation in the accumulated precipitation and land surface temperature between 2000 and 2010 on the Mongolian Plateau likely explains this finding. Substantial changes were observed in the distribution of areas with suitable habitat in 2020 relative to 2000 and 2010, and the most pronounced changes were observed in areas with medium and high suitability. The percentages of areas with high suitability on the Mongolian Plateau were 10.62% and 13.44% higher in 2020 than in 2000 and 2010, and the percentages of areas with medium suitability were 12.02% and 14.84% lower in 2020 than in 2000 and 2010, respectively. This result suggested that more habitat was suitable for the growth and development of grasshoppers in 2020 than in 2000 and 2010. The main factor mediating changes in habitat suitability for grasshoppers in 2020 relative to 2000 and 2010 was accumulated precipitation.

In our study, grasshoppers were collected from the Inner Mongolia Grassland Workstation of China. Only grasshopper occurrence data from Inner Mongolia were used (i.e., no grasshopper occurrence data from Mongolia were used); therefore, our data had certain shortcomings. The area of suitable habitat for grasshoppers in Mongolia was predicted using grasshopper occurrence records from Inner Mongolia, and the predictions were robust (AUC = 0.910). Inner Mongolia is adjacent to Mongolia and has similar climatic characteristics, and *O. decorus asiaticus* is the dominant grasshopper species in both Mongolia and Inner Mongolia. Therefore, *O. decorus asiaticus* occurrence records in Inner Mongolia can be used to predict the habitat suitability for this species in Mongolia. In addition, when we surveyed grasshopper occurrence points, the survey points may have deviated from the GPS data, which may have reduced the accuracy of the model’s assessment of grasshopper inhabitable area in the Mongolian Plateau. Topographical, soil, vegetation, and meteorological data were used in this study, but the effects of anthropogenic factors on the habitat suitability for grasshoppers, such as infrastructure construction, overgrazing, and management patterns, were not considered. These anthropogenic factors could affect the growth and development of grasshoppers by, for example, leading to the degradation of grassland and soil in the study area, which affected the availability of food and suitable habitat for grasshoppers [7,57]. Therefore, additional research is needed to facilitate the establishment of a system for controlling grasshopper outbreaks.

## 5. Conclusions

In this study, eight habitat factors, altitude, slope, grass type, soil type, vegetation coverage, aboveground biomass, accumulated precipitation, and land surface temperature, were used to simulate the distribution of suitable habitats for the *O. decorus asiaticus* on the Mongolian Plateau from 2017 to 2021 using Maxent and remote sensing data. Five key habitat factors (grassland type, accumulated precipitation, altitude, vegetation coverage, and land surface temperature) with the strongest effects on the habitat suitability for grasshoppers on the Mongolian Plateau were identified, and they were used to characterize long-term temporal and spatial changes in habitat suitability for grasshoppers on the Mongolian Plateau in 2000, 2010, and 2020. The results showed that the area of grasshopper high-suitability areas was larger in 2020 than in 2000 and 2010, mainly in the southeastern, central, and northern parts of the Mongolian Plateau; the distribution of areas with suitable habitats for grasshoppers was similar in 2000 and 2010; and the main drivers of changes in grasshopper inhabitable area in the Mongolian Plateau are accumulated precipitation and land surface temperature. The results of this study will aid the monitoring of grasshopper populations, including the vulnerability of different regions to grasshopper outbreaks, on the Mongolian Plateau in the future.

## Figures and Tables

**Figure 1 insects-14-00492-f001:**
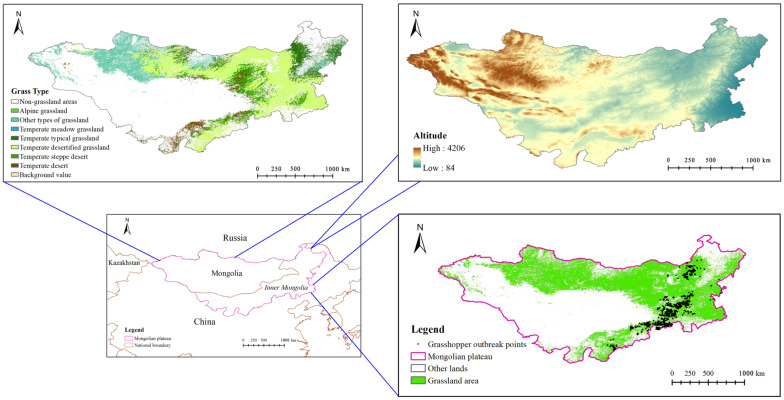
Location of the study area. The locations of *O. decorus asiaticus* outbreaks between 2018 and 2022 are indicated by the black dots. The altitude in the study area ranges from 84 m to 4206 m.

**Figure 2 insects-14-00492-f002:**
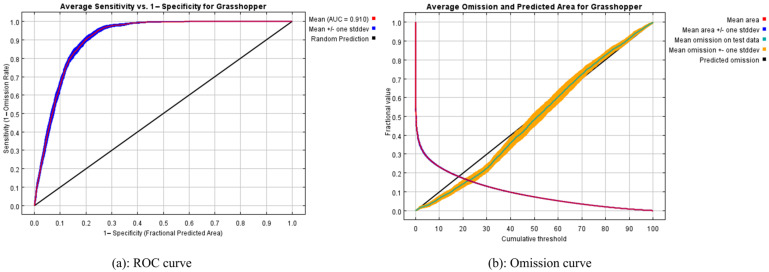
ROC curve and omission curve in Maxent models for Mongolian Plateau. The receiver operating characteristic (ROC) curve includes a red line, a blue line, and an AUC value. The red (training) line shows the fit of the model to the training data. The blue (testing) line indicates the fit of the model to the testing data. The omission curve shows the test omission rate and predicted area as a function of the cumulative threshold, averaged over the replicate runs.

**Figure 3 insects-14-00492-f003:**
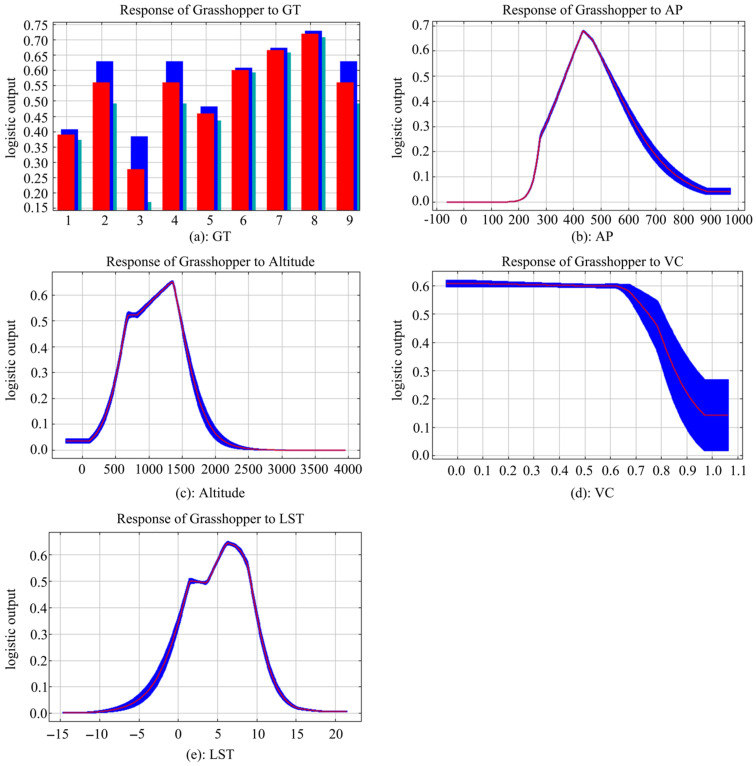
Response curves in Maxent models for key environmental variables to grasshopper occurrence. The curves show the mean response of the 15 replicate Maxent runs (red) and the mean +/− one standard deviation (blue, two shades for categorical variables). The red (training) line shows the fit of the model to the training data. The blue (testing) line indicates the fit of the model to the testing data. The histogram represents data in the format “categorical”, and the line graph represents data in the format “continuous”. GT: grass type; AP: accumulated precipitation; *VC*: vegetation coverage; LST: l and surface temperature.

**Figure 4 insects-14-00492-f004:**
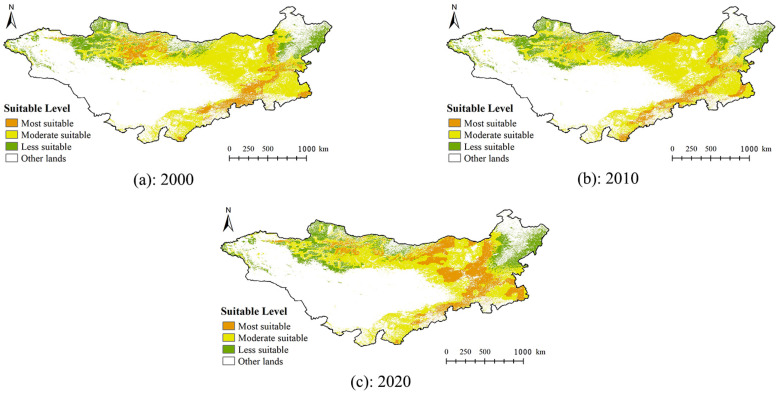
Distribution of the area of suitable habitat for grasshoppers on the Mongolian Plateau in 2000, 2010, and 2020.

**Figure 5 insects-14-00492-f005:**
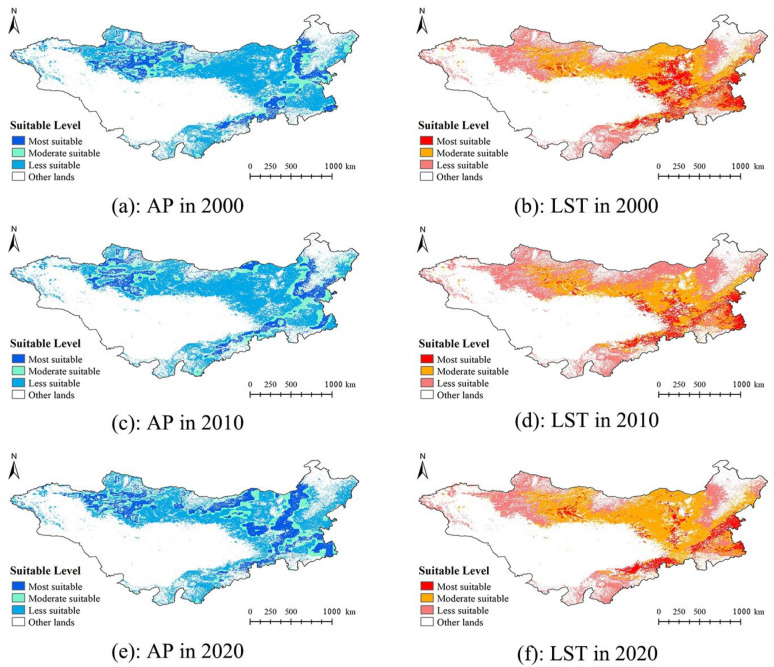
Distribution of the suitability of accumulated precipitation (AP) and land surface temperature (LST) for grasshoppers on the Mongolian Plateau in 2000, 2010, and 2020.

**Table 1 insects-14-00492-t001:** Pearson correlation analysis for each environmental variable.

	*AGB*	*VC*	ST	LST	GT	AP	Altitude	Slope
*AGB*	1.000	0.995	0.042	0.904	−0.029	0.232	0.695	0.026
*VC*	0.995	1.000	−0.017	0.911	0.000	0.241	0.714	0.009
ST	0.042	−0.017	1.000	−0.020	−0.235	−0.028	−0.094	−0.058
LST	0.904	0.911	−0.020	1.000	0.002	0.241	0.718	0.003
GT	−0.029	0.000	−0.235	0.002	1.000	−0.017	0.099	−0.126
AP	0.232	0.241	−0.028	0.241	−0.017	1.000	0.117	0.045
Altitude	0.695	0.714	−0.094	0.718	0.099	0.117	1.000	−0.426
Slope	0.026	0.009	−0.058	0.003	−0.126	0.045	−0.426	1.000

**Table 2 insects-14-00492-t002:** Environmental variables used for Maxent modeling in this study.

Category	Environmental Variables	Spatial Resolution	Data Content and Source
Topography	Altitude	1 km	Geospatial Data Cloud (https://e4ftl01.cr.usgs.gov/MEASURES/NASADEM_HGT.001/, accessed on 20 November 2022)
	Slope	1 km	
Meteorology	Land surface temperature (LST)	1 km	Mean LST: Incubation period (January–December 2017–2021, 2000, 2010, and 2020) Sources: MOD11 A2
	Accumulated precipitation (AP)	11 km (resampled to 1 km)	Mean accumulated precipitation: period (January–December 2017–2021, 2000, 2010, and 2020) Sources: ERA5-Land
Vegetation	Aboveground biomass (*AGB*)	1 km	Mean aboveground biomass: period (May–August 2017–2021, 2000, 2010, and 2020) Sources: MOD13 A2
	Vegetation coverage (*VC*)	1 km	Mean vegetation coverage: period (May–August 2017–2021, 2000, 2010, and 2020) Sources: MOD13 A2
Soil	Soil type (ST)	250 m (resampled to 1 km)	Sources: USDA (https://www.usda.gov, accessed on 20 November 2022)
Grassland	Grass type (GT)	1 km	Classification map of grassland in Eurasia (2009) Sources: National Tibetan Plateau Data Center (http://data.tpdc.ac.cn, accessed on 20 November 2022)

**Table 3 insects-14-00492-t003:** Classification of the habitat suitability for grasshoppers on the Mongolian Plateau.

Suitability Grade	Inhabitability Index (*IH*)	Description of Area
Level 1 (high)	*IH* ≥ 2.5	Highly suitable for grasshopper growth and development; high risk of experiencing damage from grasshoppers
Level 2 (middle)	1.5 ≤ *IH* < 2.5	Suitable for grasshopper growth and development; medium risk of experiencing damage from grasshoppers
Level 3 (low)	1.0 ≤ *IH* < 1.5	Low suitability for grasshoppers; low risk of experiencing damage from grasshoppers

**Table 4 insects-14-00492-t004:** Relative contributions of the environmental variables to grasshopper occurrence.

Environmental Variables	Percentage Contribution	Permutation Importance
GT	49.0%	4.3%
AP	23.8%	58.3%
Altitude	12.4%	16.2%
*VC*	6.3%	2.4%
LST	4.0%	13.4%
ST	3.2%	1.9%
Slope	1.2%	1.6%
*AGB*	0.1%	1.9%
Total	100%	100%

GT: grass type; AP: accumulated precipitation; *VC*: vegetation coverage; LST: land surface temperature; ST: soil type; *AGB*: aboveground biomass.

**Table 5 insects-14-00492-t005:** Relative contributions of the key environmental variables to grasshopper occurrence.

Environmental Variables	Percentage Contribution
GT	51.3%
AP	24.9%
Altitude	13.0%
*VC*	6.6%
LST	4.2%
Total	100%

**Table 6 insects-14-00492-t006:** Classification results of the key environmental variables.

Environmental Variables	Low	Moderate	High
GT	1, 3, 5	2, 4, 9	6, 7, 8
AP (mm)	<369 mm, >533 mm	369–404 mm, 489–533 mm	404–489 mm
Altitude (m)	<664 m, >1480 m	664–1131 m, 1404–1480 m	1131–1404 m
*VC*	>0.75	0.59–0.75	<0.59
LST (°C)	<1.56, >9.19	1.56–5.38, 8.17–9.19	5.38–8.17

GT: 1, non-grassland areas; 2, alpine grassland; 3, other types of grassland; 4, temperate meadow grassland; 5, temperate typical grassland; 6, temperate desertified grassland; 7, temperate steppe desert; 8, temperate desert; and 9, the background value.

## Data Availability

The data are not publicly available because the data need to be used in future work.

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
