# Peer review of "Analysis of Spatiotemporal Variation in Habitat Suitability for Oedaleus decorus asiaticus Bei-Bienko on the Mongolian Plateau Using Maxent and Multi-Source Remote Sensing Data"

_insects, 2023, doi:10.3390/insects14060492_

Round 1

Reviewer 1 Report

All suggestions are in the attached file.

Author Response

Response to Reviewer 1's comments

Thank you very much for your comments and suggestions. Your useful feedback has helped us to improve our paper substantially. We hope our responses address your concerns.

  • Line 3. Add taxonomic classifier.

[Response] We have changed all " Oedaleus decorus asiaticus" to " Oedaleus decorus asiaticus Bei-Bienko" in the article.

  • Line 25. What kind of data? What kind of variables, how many?

[Response] We have reviseded the sentence to " In this study, spatiotemporal variation in habitat suitability for Oedaleus decorus asiaticus Bei-Bienko on the Mongolian Plateau was assessed using maximum entropy (Maxent) modeling along with multi-source remote sensing data (meteorology, vegetation, soil and topography) ".

  • Line 28. Add a sintex of the methodology used, if you only list the results may look confusing.

[Response] We have reviseded the sentence to " Based on the assessment results of suitability by maxent model, the model threshold settings and the formula for calculating the inhabitability index, the 2000s, 2010s and 2020s inhabitable area were calculated. The results show that ".

  • Line 45. Add a reference for this information.

[Response] We have added a reference.

  • Line 57. Add reference for this information. Looks confusig...With satellite data you can identify the optimal areas for grasshopers, it is not really important the size of the organism in study. The objective is to find out optimal areas, not the size of the organism, aunque es posible calcular tamaños estimados y numero de generacion a partir de información satellite.

[Response] We have added a reference. We mean that due to the small size of grasshoppers, it is not possible to identify the grasshopper distribution by satellite images alone. Therefore, remote sensing technology is needed to obtain data on habitat factors affecting grasshopper growth and then to invert the grasshopper distribution.

  • Line 91. It is just one model, there are other, but are not mention in the text, if you referred to SDMs (Species Distribution Models) mention them.

[Response] This study is based on the maximum entropy model, which has been described in the previous section on ecological niche models. And is highlighted here to show that the maximum entropy model is widely used and suitable for assessing and predicting the distribution of suitable habitats for grasshoppers.

  • Line 92. Replace “Maxent theory”, with: “Maximum entropy theory”

[Response] We have revised " Maxent theory " to " Maximum entropy theory ".

  • Line 102. This belongs to the methodology section.

[Response] We have reviseded the sentence to " Here, the integration of the Maxent model and remote sensing data is used to conduct a species-environmental matching model. This study aims to: (1) Determine the distribution of grasshopper’s inhabitable area on the Mongolian plateau; (2) Identify key environmental factors affecting the distribution of grasshopper on the Mongolian plateau; (3) Analyze the spatial and temporal evolution of grasshoppers inhabitable area on the Mongolian plateau in 2000, 2010 and 2020, and identify the main driving factors leading to changes in grasshopper inhabitable area on the Mongolian plateau. The results of this study enhance our understanding of the distribution of grasshoppers on the Mongolian plateau and will help identify regions on the Mongolian Plateau that will be most vulnerable to grasshopper outbreaks in the future."

  • Line 134. It would be more useful if you add a map for every outbreak point for year, or identified every year outbreak point with a different color. This information permitira ver si estos puntos se presentaron en los mismos años o ha cambiado la tendencia.

[Response] Thank you for your valuable suggestions. In this study, we surveyed the grasshopper occurrence sites by route survey, and did not obtain grasshopper occurrence at fixed sites. Therefore, we believe that it is difficult to tell the trend of grasshopper occurrence over the years from the information obtained from the grasshopper occurrence sites. In order to improve the basic information of the study area, we supplemented the altitude and grass type distribution map of the Mongolian Plateau in Figure 1.

  • Line 140. If you have the number of surveys for every station, add it ...

[Response] Thank you for your valuable comments. Since our grasshopper occurrence sites are obtained through route surveys. Therefore, number of grasshopper surveys without fixed survey stations.

  • Line 143. If you referred to presence point, it would be better to specifie as grasshopper presence data or grasshopper occurrence.

[Response] It is grasshopper occurrence point. We have revised " grasshopper points " to " grasshopper occurrence point ".

  • Line 160.Did you explore multicollinearity into the variables used? This may affect the results…

[Response] We did Pearson correlation analysis for each environmental variable. We have added the sentence as "To avoid strong collinearity between variables, we did a Pearson correlation analysis for each variable. The statistical analysis was implemented in SPSS Statistics 22.0 (IBM Corp, Armonk, NY, USA). The results of the analysis showed that vegetation coverage was strongly correlated with above ground biomass. However, the purpose of our simulation by maxent was to evaluate the weights of each habitat factor of grasshopper, and in the maxent model, the jackknife would have differentiated the contribution of the two indicators, which attenuated the effect of strong correlation on the model results. Meanwhile, since overgrazing is more common in the study area, this further reduces the correlation between the variables."

  • Line 169-172. The spatial resolution of the information should be the same, otherwise, methods need to be used to adjust the information to the same resolution.I recommend reviewing the information from WorldClim or Chelsea

[Response] We use the resampling function in the ArcGIS software to resample the spatial resolution of all environmental variables to 1km.

  • Line 169-172. What is the purpose of using this information, and the selected periods?

[Response] The purpose of using these environmental variables was to evaluate the weights of each environmental variable through the maxent model. The selected periods correspond to the various growth stages of the grasshopper, which includes the spawning period (July-September), overwintering period (November-March), hatching period (April-May), and growing period (June). The influence of each habitat factor on grasshopper growth and development is described in subsection 2.2.2. of the article.

  • Line 169-172. Already mentioned, what is the purpose of using this data?

[Response] The purpose of using these environmental variables was to evaluate the weights of each environmental variable through the maxent model. The selected periods correspond to the various growth stages of the grasshopper, which includes the spawning period (July-September), overwintering period (November-March), hatching period (April-May), and growing period (June). The influence of each habitat factor on grasshopper growth and development is described in subsection 2.2.2. of the article.

  • Line 211. What method did you use to match all the information?

[Response] Using the resampling function, cropping function and raster to ASC function in ArcGIS software, all data are unified in coordinate system, data format, raster size, row number and processing range operation, so that the spatial resolution of all data is unified to 1km, and finally the data in raster form are converted to ASCII files. All these processes are unified through the built-in Python plug-in of ArcGIS.

  • Line 220.replicates?

[Response] The model randomly selects 70% of the data as training data and 30% of the data as test data and runs 15 times repeatedly. The points selected each time are randomized, and the result is taken as the average of the 15 simulations.

  • Line 221. Was the resulting models an ensamble?

[Response] The resulting model is an ensemble. The model randomly selects 70% of the data as training data and 30% of the data as test data and runs 15 times repeatedly. The points selected each time are randomized, and the result is taken as the average of the 15 simulations.

  • Line 239.What about logistic treshold considering training omission rate, test omission rate, P-value?

[Response] Logistic threshold is 0.053; Training omission rate is 0.005; Test omission rate is 0.011; P-value is 0.05. All results are the average of 15 randomly repeated runs.

  • Line 239. Add reference.

[Response] We have added two references.

  • Line 245.Based on what facts? Add information that helps to understand the proposed classification.

[Response] We have reviseded the sentence to "Based on the literature and expert experience". We have added two references.

  • Line 282.This information is the average of the 15 replicates? Add the meaning of every environmental variable:GT= AP=.

[Response] The run results are the average of 15 randomly repeated runs. The meaning of each environment variable is added at the bottom of Table3.

  • Line 308.This is not very clear since the first part, be more precise while explaining all the methodology.

[Response] We have reviseded the sentence to " Based on the assessment results of suitability by maxent model, the model threshold settings and the formula for calculating the inhabitability index, the 2000s, 2010s and 2020s inhabitable area were calculated (Fig. 4). The results show that the areas with suitable habitat for grasshoppers on the Mongolian Plateau were similar in 2000 and 2010."

  • Line 329-330. This information is valuable, but it will be more interesting to add data for the future. This would help to plan efficient strategies.

[Response] Thank you for your valuable comments. In this study, our main objective is to extract the grasshopper’s inhabitable area on the Mongolian Plateau in 2000, 2010 and 2020, and then analyze the spatial and temporal evolution of the grasshopper’s inhabitable area on the Mongolian Plateau over the past 20 years to understand the main drivers of the change in the inhabitable area. The study can provide guidance for future monitoring and early warning of grasshopper on the Mongolian plateau. For example, in the future, changes in the main drivers can be monitored to predict grasshopper occurrence, and thus prevent and control measures can be taken in advance. We will take your suggestions into consideration in the next research.

Once again we thank you for your all comments and suggests and we hope you will find our revision satisfactory for publication in Insects.

Author Response

Response to Reviewer 2's comments

Thank you very much for your comments and suggestions. Your useful feedback has helped us to improve our paper substantially. We hope our responses address your concerns.

  • Line 13. Oedaleus decorus asiaticus” should be written in italic

[Response] We have changed all " Oedaleus decorus asiaticus" to " Oedaleus decorus asiaticus Bei-Bienko" in the article.

  • Line 21. Please refine the scientific names. Please be consistent throughout the manuscript.

[Response] We have changed all " Oedaleus decorus asiaticus" to " Oedaleus decorus asiaticus Bei-Bienko" in the article.

  • Introduction does not provide sufficient review of the target species in the context of this study particularly at regional or global scale. - Please clearly outline the objectives of the study.

[Response] In the third paragraph of the introduction, we added " Oedaleus decorus asiaticus Bei-Bienko belongs to the family Acrididae; it is one of the most dominant grasshopper species on the Mongolian Plateau. This species induces major damage to Leymus chinensis, Stipa, and other plants. Outbreaks of Oedaleus decorus asiaticus Bei-Bienko can lead to rapid reductions in the area of pasture and grassland degradation. Oedaleus decorus asiaticus Bei-Bienko has become an indicator species of grassland degradation. Controlling Oedaleus decorus asiaticus Bei-Bienko becomes extremely difficult once swarms begin to engage in collective behavior because of their long-distance dispersal capacities, destructive capabilities, and explosive population growth. Characterizing the habitat suitability for this grasshopper species on the Mongolian Plateau is thus critically important for developing methods to minimize the impacts of future outbreaks."

  • Line 145. It is not clear what is the minimum distance between to sample points. And which sampling technique or strategy was followed is not clear? Please clarify and back it up with references.

[Response] We have reviseded the sentence to "A regional survey method according to the standard in agricultural industry of People’s Republic of China (NY/T 1578-2007, Rules for investing locality and grasshopper in grassland) was used to investigate the overall grasshopper occurrence. A total of 1,745 grasshopper occurrence points were obtained by route survey method for spring, summer and autumn seasons from 2018 to 2022, and the distance between sampling points was not less than 100m. The survey routes covered all major geomorphic units and areas vulnerable to grasshopper outbreaks."

  • Line 167. section “2.2.2.1. Remote sensing data” this section can be merged with the section “2.2.2” please.

[Response] Thank you for your comments, we have adjusted the paragraphs in accordance with your suggestions.

  • Line 185.section “2.2.2.2. Other geospatial data” this section can be merged with the section “2.2.2” please.

[Response] Thank you for your comments, we have adjusted the paragraphs in accordance with your suggestions.

  • Line 210. It is not clear how the issue of multicollinearity is dealt with between the environmental variables. Please clarify accordingly.

[Response] We add a description of the correlation between variables in subsection 2.2.2. "To avoid strong collinearity between variables, variables with Pearson correlation coefficients less than 0.9 were retained. The statistical analysis was implemented in SPSS Statistics 22.0 (IBM Corp, Armonk, NY, USA). The results of the analysis showed that vegetation coverage was strongly correlated with above ground biomass. However, the purpose of our simulation by maxent was to evaluate the weights of each habitat factor of grasshopper, and in the maxent model, the jackknife would have differentiated the contribution of the two indicators, which attenuated the effect of strong correlation on the model results. Meanwhile, since overgrazing is more common in the study area, this further reduces the correlation between the variables. "

  • Line 221. The default values were used for the remaining parameters.” Default mode does not necessarily generate accurate outputs? For example, Default mode sets max number of background points as 10,000” the presence records are (n= 1745) ? a ratio of 1745:10000 ? Less background points should have been examined in order to see other results?

[Response] We agree with your comments and have modified the model settings accordingly. Our model settings are not all default settings, for example, the regularization parameters are the result of filtering after comparing multiple combinations. Also, we adjusted the number of background points, but this did not affect the results.

  • Line 223. "AUC" Sometimes AUC alone is not sufficient to evaluate the model performance. Why TSS was not considered alongside the AUC?

[Response] Thank you for your valuable comments. Based on your comments, we added a TSS test to the model, and the results of 15 runs of the model showed that the TSS was 0.631, and the TSS test results showed that the model performance was fair. We have reviseded the sentence to " The accuracy of the Maxent model was evaluated using the area under curve (AUC) of the receiver operating characteristic (ROC) curve and TSS."

  • Results - The captions for the majority of the Figures and Tables are not well described. Please describe them for better clarity.

[Response] Thank you for your valuable comments. We have reviseded the captions to " Figure 2. ROC curve and Omission curve in Maxent models for Mongolian Plateau. The receiver operating characteristic (ROC) curve includes a red line and blue line and an AUC value. The red (training) line shows the fit of the model to the training data. The blue (testing) line indicates the fit of the model to the testing data. Omission curve shows the test omission rate and predicted area as a function of the cumulative threshold, averaged over the replicate runs.", " Table 3. Relative contributions of the environmental variables to grasshopper occurrence", " Table 4. Relative contributions of the key environmental variables to grasshopper occurrence", " Figure 3. Response curves in Maxent models for key environmental variables to grasshopper occurrence. The curves show the mean response of the 15 replicate Maxent runs (red) and and the mean +/- one standard deviation (blue, two shades for categorical variables). The red (training) line shows the fit of the model to the training data. The blue (testing) line indicates the fit of the model to the testing data.", and " Table 5. Classification results of the key environmental variables"

  • Line 374-383: This paragraph is redundant. Please focus on discussing the results and draw parallel with previous studies. -In depth explanations of how exactly environmental variables influence the spatial distribution of the species is not sufficient from the discussion. -Furthermore, the discussion should also highlight the benefit and limitations of the applied modeling techniques particularly when it comes to the implications of the current techniques in management and precautionary efforts.

[Response] We have reviseded the paragraphs to " Maxent models are based on Maxent theory, and they can be used to predict the potential distribution of species through occurrence records and environmental variables. Maxent models simplify complex natural systems and permit reliable predictions to be obtained with small sample sizes. The results of the maxent model simulation showed that the average AUC value for 15 repeated runs was 0.910 and the TSS value was 0.631. Meanwhile, the average omission rate of the test sample was close to the predicted omission rate as shown by the results of the omission curve. The above results indicate that the model has a good performance. However, since grasshoppers are an outbreak species, this may reduce the accuracy of the model. In addition, since our model is for Oedaleus decorus asiaticus Bei-Bienko and is limited to the spatial extent of the Mongolian plateau, there is a real limitation in model extrapolation, and only a small range of model extrapolation can be performed, although clamping is possible."

Once again we thank you for your all comments and suggests and we hope you will find our revision satisfactory for publication in Insects.

Author Response

Response to Reviewer 3's comments

Thank you very much for your comments and suggestions. Your useful feedback has helped us to improve our paper substantially. We hope our responses address your concerns.

  • L42: Repetition of deleterious impacts on animal husbandry and agriculture seems unnecessary given this was mentioned a few lines earlier (L38-39)?

[Response] Thank you for your valuable suggestions. We have reviseded the sentence to " Grasshoppers can have major deleterious effects on the agriculture and animal husbandry industries because of their explosive growth, destructive capabilities, and migratory behavior. In addition, outbreaks of grasshoppers can induce major harm to the environment and human society, such as grassland degradation and desertification."

  • Line 43-46. These statements require appropriate citations.

[Response] We have added a reference.

  • Line 48. Also requires a citation.

[Response] We have added a reference.

  • Line 50-53. Repetitive and requires an appropriate citation. The authors need to add any supporting literature to contextualise what is currently known of the grasshoppers in the study region here. E.g. Are there dominant grasshopper species? Are these grasshoppers native species or invasive?

[Response] Thank you for your valuable suggestions. We have added a reference. There are few studies on Oedaleus decorus asiaticus Bei-Bienko in the Mongolian Plateau, and only a few studies have been conducted in Inner Mongolia, China. Oedaleus decorus asiaticus Bei-Bienko is the dominant locust species in the Mongolian plateau region and is a native species. We have added content about Oedaleus decorus asiaticus Bei-Bienko in the Mongolian Plateau. " Oedaleus decorus asiaticus Bei-Bienko belongs to the family Acrididae; it is one of the most dominant grasshopper species on the Mongolian Plateau. This species induces major damage to Leymus chinensis, Stipa, and other plants. Outbreaks of Oedaleus decorus asiaticus Bei-Bienko can lead to rapid reductions in the area of pasture and grassland degradation. Oedaleus decorus asiaticus Bei-Bienko has become an indicator species of grassland degradation. Controlling Oedaleus decorus asiaticus Bei-Bienko becomes extremely difficult once swarms begin to engage in collective behavior because of their long-distance dispersal capacities, destructive capabilities, and explosive population growth. Characterizing the habitat suitability for this grasshopper species on the Mongolian Plateau is thus critically important for developing methods to minimize the impacts of future outbreaks."

  • Line 53-56. It would be valuable to indicate how the monitoring programme would be improved based on knowledge of predicted species distributions, and how this information could feed into development of appropriate management programmes? For example, one sentence indicating that knowledge of the predicted distributions of the grasshoppers could identify high-priority regions to conduct surveys and dispatch rapid response teams to manage outbreaks?

[Response] Thank you for your valuable suggestions. We have reviseded the sentence to "Therefore, it is necessary to use multi-source remote sensing data to invert the distribution of grasshoppers and predict the grasshopper outbreak areas in order to formulate countermeasures in advance to prevent the occurrence of grasshoppers and reduce the damage of grasshoppers to the local ecological environment and social economy."

  • Line 64-66. Change "grasshop-pers" to "grasshoppers"

[Response] We have changed " grasshop-pers " to "grasshoppers".

  • Line 76. What is meant by manually? Sweep nets, hand-picking, traps? Please clarify.

[Response] It is hand-picking. Traditional grasshopper assessment methods require manual field surveys to assess the likelihood of grasshopper outbreaks in the area, which is time-consuming and labor-intensive. However, the inversion of the grasshopper potential areas by remote sensing technology will greatly reduce the human and material costs and is more efficient.

  • Line 76-85. This paragraph is unclear to me. At the start of the previous paragraph, the authors stated that "The distributions of grasshoppers cannot be characterized using satellite remote 57 sensing images because of their small size. (L57-58)", but now the authors state that remote sensing can be used to monitor grasshoppers? Please clarify.

[Response] We have added a reference. We mean that due to the small size of grasshoppers, it is not possible to identify the grasshopper distribution by satellite images alone. Therefore, remote sensing technology is needed to obtain data on habitat factors affecting grasshopper growth and then to invert the grasshopper distribution.

  • Line 99. On first mention of the grasshopper species, please provide the species authority, order and family name.

[Response] Thank you for your valuable suggestions. We added the sentence "Oedaleus decorus asiaticus Bei-Bienko belongs to the family Oedipodidae,order Orthoptera."

  • Line 105-108. The current study models the distribution of one grasshopper species (Oedaleus decorus asiaticus). As such, I think the authors need to temper the proposed wide reaching implications of their study and focus on the implications for the surveillance and management of the study species.

[Response] Thank you for your valuable comment. We have reviseded the sentence to " This study aims to: (1) Determine the distribution of Oedaleus decorus asiaticus Bei-Bienko inhabitable area on the Mongolian plateau; (2) Identify key environmental factors affecting the distribution of Oedaleus decorus asiaticus Bei-Bienko on the Mongolian plateau; (3) Analyze the spatial and temporal evolution of Oedaleus decorus asiaticus Bei-Bienko inhabitable area on the Mongolian plateau in 2000, 2010 and 2020, and identify the main driving factors leading to changes in Oedaleus decorus asiaticus Bei-Bienko inhabitable area on the Mongolian plateau. The results of this study enhance our understanding of grasshopper distribution on the Mongolian plateau and will provide guidance for monitoring and management of grasshoppers on the Mongolian plateau."

  • Line 112-122. The section discussing the geography and climate of the Mongolian Plateau requires some citations supporting the classification of the area.

[Response] We have added two references.

  • Line 121. Insert a paragraph break at the end of this line so separate the discussion about the geography/climate of the region and the focal taxon. Figure 1: A slightly more informative map would improve the quality of the paper. For example, plotting the major biomes or climatic zones (e.g. the Koppen-Geiger zones) across the plateau as the base layer for Figure 1 would provide the reader with much needed context.

[Response] Thank you for your valuable comment. In order to improve the basic information of the study area, we supplemented the altitude and grass type distribution map of the Mongolian Plateau in Figure 1.

  • Line 138-142. Were grasshoppers recorded at all 1745 survey points? Please clarify. Are there any GPS points where the grasshoppers were not recorded (i.e. absence points)? Were any online repositories (e.g. GBIF) checked for additional GPS points? If not, please explain why? The more complete the understanding and representation of the distribution of the species being modelled in MaxEnt, the more realistic the outputs generated.

[Response] Yes, grasshoppers were recorded at all 1745 points. We did not record points where grasshoppers did not occur. Since the Oedaleus decorus asiaticus Bei-Bienko is endemic to the Mongolian plateau, all point data were accumulated by our route surveys in recent years, and the online database does not have corresponding records.

  • Line 141-142. The survey points and GPS data may show a bias due to the sampling being performed in regions that are already known to be vulnerable to grasshopper outbreaks. If you expected to find grasshoppers in this region, you have effectively sampled high-quality habitat. But, this may lead to an underestimate of suitable habitat for the grasshopper because marginal habitats have not been sampled. This bias and its potential implications needs to be acknowledged in the manuscript

[Response] Thank you for your comments, we acknowledge that there may be bias in the survey points and GPS data, and we will explain the existence of this bias in the discussion section. We added " In addition, when we surveyed grasshopper occurrence points, the survey points may deviate from the GPS data, which may reduce the accuracy of the model's assessment of grasshopper inhabitable area in the Mongolian Plateau." to the discussion section. In the subsequent study, we will continue to improve the method of obtaining locust occurrence points to reduce the errors in the study.

  • Line 149. The predictor/environmental variable selection is generally well described. The lack of an explicit variable quantifying overwintering temperatures seems like it may be important omission from the model building process. The mean LST data layer is likely to broad to capture important periods in the grasshoppers biology that may be affected by climate. I would strongly advise that the authors calculate a metric such as the mean lowest temperatures in the coldest month or quarter of the year (I assume this would be quite straightforward from the LST data?) and use this as an additional predictor in the MaxEnt model.

[Response] Thank you for your valuable comment. In this study, our main objective was to extract the inhabitable area of the grasshopper in the Mongolian plateau, which is a macroscopic study and not limited to one stage of the grasshopper. In the future, we will build on this macroscopic study to examine the risk of occurrence of grasshopper at each reproductive stage. At that time, we will consider the environmental variables that affect the development of the grasshopper at each reproductive stage (like the overwintering temperature variable you mentioned). Thank you again for your valuable comment on our study.

  • Line 157-159. I assume that the nymph and emergence period is May-August given that the vegetation layers in Table 1 are calculated for this time period? If so, please clarify this in the text, otherwise it isn't clear why the vegetation data is only quantified for this short time period.

[Response] We have reviseded the sentence to " The life cycle of Oedaleus decorus asiaticus Bei-Bienko includes the spawning period (July-September), overwintering period (November-March), hatching period (April-May), growing period (June). Among them, nymph and emergence period are in May to August."

  • Line 202-209. The formula provided seems unnecessary. A simple explanation of how MaxEnt works by correlating species GPS points with environmental layers, as the authors have already provided, is sufficient.

[Response] Thank you for your valuable comment. The formula is provided to explain the principle of the maximum entropy model more intuitively.

  • Line 217. An explanation of what 'feature classes' are, and how they are used in MaxEnt model calibration, should be included here.

[Response] 'feature classes' refers to the data type of each environment variables, including continuous numeric type, category type, etc. The grass type and soil type in this study are categorical data, which are set to "Categorical" in the maxent model, while the other data are continuous data, which are set to "Continuous" in the maxent model.

  • Line 217. Do you mean that seperate MaxEnt models were calibrated with only Hinge, then only quadratic, then only Linear features and so on? Please clarify what combinations of feature classes were tested.

[Response] We use a combination of linear/quadratic/product, categorical, threshold, hinge features.

  • Line 221-222. If optimal feature classes were selected (see my comment above on L217), why were default values for the regularisation multiplier used? It is commonplace to optimise the feature class and regularisation multiplier values used for MaxEnt models, and has been shown to significantly improve model fit in many cases (see Merow et al., 2013 and references therein).

[Response] We agree with the comment you made and have made changes accordingly. We have reviseded the sentence to " Model training methods used included the linear, quadratic, hinge, product, and auto programs, our model compares several combinations of regularization parameters and selects the best ones. "

  • Line 220. The model evaluation process may be biased by the lack of independent testing data (i.e. the withheld 30%). Ideally, to evaluate model accuracy, the testing data should be independent of the training data. This is not the case in the current modelling workflow. When the test and training data are not independent, the assessment of model accuracy is not adequate. The simplest solution to this problem is to perform spatial cross-validation, whereby the testing and training data are split into folds (e.g. 70% / 30% test, as the authors have already done), however, the data are spatially grouped prior to the data splitting to create spatial independence (at least artificial independence) between the test and training data. This is not ideal either as these splits are not truly independent, however, spatial cross-validation provides a better indication of model accuracy than the method currently used. Otherwise, the models may be overfitting to significantl, which may bias model outputs and provide false confidence in model performance. Overfitting refers to the situation where the model performs very well internally (i.e. when tested on the same data used to build the model, like the current study). However, when the model is projected into novel geographic/climatic regions, the model performs poorly. Spatial cross-validation at least partially protects against significant model overfitting by allowing the user to assess model accuracy on novel GPS/climatic data per fold/run.

[Response] We agree with the comment you made and have made changes accordingly. We use 70% of the points for model simulation and 30% of the points for model testing. These points are selected randomly and uniformly by the model, which ensures the spatial independence to some extent. We adopt the method of repeating 15 simulations to take the average value, and each simulation is randomly selected for simulation and testing points, which also reduces the problem of model overfitting. We have reviseded the sentence to " During each run, 70% of the occurrence records were used as the training set, and 30% of the occurrence records were used as the test set, and these points are selected randomly and uniformly by the model, which can ensure spatial independence to some extent. "

  • Line 223. The methodology provided for fitting the SDM models are insufficient and it would not allow for the models to be replicated. If I were given the GPS data by the authors, I would not be able to reproduce their models due to a lack of supporting information. Please consult some recent published literature that uses the same modelling algorithms and read the methods section to see what other information is typically required when discussing the methodology. E.g. Was model extrapolation allowed? If so, was any clamping allowed, and how was clamping implemented?

[Response] Thank you for your valuable comment. We have added the appropriate notes in the discussion section. We added " In addition, since our model is for Oedaleus decorus asiaticus Bei-Bienko and is limited to the spatial extent of the Mongolian plateau, there is a real limitation in model extrapolation, and only a small range of model extrapolation can be performed, although clamping is possible." to the discussion section.

  • Line 223. Following on from my above point, a critical component of most SDM studies is the acquisition of absence GPS points or the generation of pseudo-absences to test model performance. I assume that no true absence data were available and the authors generated pseudo-absence data? Please clarify if and how (pseudo)-absence points were generated? How was the background area defined? How many points were generated? Two papers are listed below that could be helpful. Barber, R. A., Ball, S. G., Morris, R. K., & Gilbert, F. (2022). Target‐group backgrounds prove effective at correcting sampling bias in Maxent models. Diversity and Distributions, 28(1), 128-141. Merow, C., Smith, M. J., & Silander Jr, J. A. (2013). A practical guide to MaxEnt for modeling species' distributions: what it does, and why inputs and settings matter. Ecography, 36(10), 1058-1069.

[Response] We recorded grasshoppers at all 1745 points and did not record grasshopper non-occurrence points. The model background points were defined by the 1745 points where grasshopper occurrence was recorded.

  • Line 223-225. The AUC is not a suitable metric for most MaxEnt studies due to issues with interpretation of the statistic when used with pseudo-absence data versus true presence/absence data. It is unclear whether the authors used absence data to evaluate their MaxEnt models, which has been brought up previously in my review. This response to this query has implications for whether AUC is valid or not (see Lobo et al., 2008; Warren et al., 2019 and references therein). If pseudo-absences have been used, then the authors need to adopt another suite of metrics to evaluate model performance that are appropriate for the underlying data (e.g. omission rates, AICc, pROC, ect...)( Lobo, J. M., Jimenez-Valverde, A., & Real, R. (2008). AUC: A misleading measure of the performance of predictive distribution models. Global Ecology and Biogeography, 17(2), 145– 151. https://doi.org/10.1111/j.1466-8238.2007.00358.x)(Warren, DL, Matzke, NJ, Iglesias, TL. Evaluating presence-only species distribution models with discrimination accuracy is uninformative for many applications. _J Biogeography. 2020; 47: 167– 180. https://doi.org/10.1111/jbi.13705)

[Response] Thank you for your valuable suggestions. We have added omission curves and TSS. We have reviseded the sentence to " The accuracy of the Maxent model was evaluated using the area under curve (AUC) of the receiver operating characteristic (ROC) curve, omission curves and TSS. The value of the AUC ranges from 0 and 1. The closer the values of AUC and TSS are to 1, the higher the accuracy of the model. The test omission in the omission curve is close to the predicted omission, the better the simulation result is indicated."

  • Line 225. An assessment of model extrapolation is required to indicate the presence and magnitude of extrapolation versus interpolation for the SDM models. When projecting models into novel geographic regions, time periods and/or climatic conditions, there is a risk that the outputs obtained will be biased by extrapolation, and produce nonsensical/implausible predictions. While some extrapolation is usually unavoidable , it is generally best-practise to assess where and to what extent models are extrapolating, and temper any inferences made in regions in extrapolation. For example, Multi-surface Environmental Similarity Surfaces (MESS) could be computed (Elith et al., 2011). Please see the below reference for further information. Elith, J., Phillips, S. J., Hastie, T., Dudík, M., Chee, Y. E., & Yates, C. J. (2011). A statistical explanation of MaxEnt for ecologists. Diversity and distributions, 17(1), 43-57.

[Response] Thank you for your valuable comment. We have added the appropriate notes in the discussion section. We added " In addition, since our model is for Oedaleus decorus asiaticus Bei-Bienko and is limited to the spatial extent of the Mongolian plateau, there is a real limitation in model extrapolation, and only a small range of model extrapolation can be performed, although clamping is possible." to the discussion section.

  • Line 231-240. It is not sufficient to just mention that you consulted the literature. Please provide appropriate citations for the methodology being used. The derivation of the suitability index is a critical component of the study and it needs to be adequately cited and verified. Currently, there is no justification for any of the thresholds and values being used, which needs to be rectified. I am not sure that I see the value of deriving this suitability index when there are a variety of indices already available in the SDM literature, and which are frequently used in MaxEnt, to transform the raw MaxEnt output into a binary map (e.g. Liu et al., 2016)? Liu C, Newell G, White M. 2016. On the selection of thresholds for predicting species occurrence with presence-only data. Ecology and Evolution 29;6(1):337-48. doi:10.1002/ece3.1878.

[Response] Combining literature and expert experience. We have added four references.

  • Line 259. Is the "0.192" supposed to be "0.912"? L259: As per my previous comments, the model evaluation process needs to be updated using spatial cross-validation and models need to be evaluated using an appropriate suite of metrics (not AUC, assuming that no true absences were available). L296: Am I misinterpreting the accumulated precipitation values, but in the methods, the total annual rainfall was quoted as 200mm per annum, but now accumulated rainfall is > 5000mm? Please clarify.

[Response] Sorry, we made a mistake when writing the article, we have made changes where appropriate. the minimum value of AUC is 0.912. We have added omission curves and TSS. We wrote the wrong value for the accumulated precipitation and have made changes in the corresponding place.

  • Line 368. Please elaborate on what are these 'other factors'?

[Response] 'other factors' refers to the migratory characteristics of grasshoppers. We have reviseded the sentence to " Acquiring field survey data of grasshoppers in the study region is difficult because of the complexity of the terrain and migratory characteristics of grasshoppers. "

  • Line 376. They can be used with small sample sizes, this doesn't mean that they perform well. Careful attention needs to be paid to data availability and quality and model performance metrics to assess how well the models may be performing.

[Response] Thank you for your valuable suggestions. We have added omission curves and TSS. The results of AUC, TSS and omission curves show that the performance of the model is fair.

  • Line 377. See my previous comment about the use of AUC-ROC to assess within-sample model performance. This metric is usually not suitable for assessing MaxEnt model performance and needs to be updated with a more appropriate suite of model evaluation metrics and model evaluation techniques (e.g. spatial cross validation).

[Response] Thank you for your valuable suggestions. Since the test and training data are randomly selected by the model and the data are independent of each other, this attenuates the spatial crossover phenomenon to some extent. In addition, we add omission curves and TSS to the model to check the performance of the model, the results of AUC, TSS and omission curves show that the performance of the model is fair.

  • Line 396-397. Discuss these previous studies and provide appropriate citations. What possible reasons may explain the discrepancy between the current study and previous studies?

[Response] We have added a reference. We have made changes to the content of the paragraphs. " The cumulative percentage contribution of grass type, accumulated precipitation, altitude, and vegetation coverage was 91.5%. The percentage contribution of land surface temperature was only 4%, but its importance was high. This might stem from the temperate continental climate of the Mongolian Plateau. There is a large temperature difference between winter and summer; extreme cold weather can occur in winter, and low temperatures hinder the growth and development of grasshoppers and the hatching of grasshopper eggs [27]. Some studies have shown that soil moisture and temperature have a greater effect on grasshoppers during the egg-laying, hatching, and cyst stages [66]. Therefore, grass type, accumulated precipitation, altitude, vegetation coverage, and land surface temperature were the key habitat factors determining the suitability of habitat for grasshoppers on the Mongolian Plateau. The effects of soil type, slope, and aboveground biomass on the habitat suitability for grasshoppers on the Mongolian Plateau were weaker than the five aforementioned key habitat factors."

  • Line 420-422. The AUC does not allow you to infer the performance of the model in novel regions.

[Response] Thank you for your valuable suggestions. We have added omission curves and TSS. The results of AUC, TSS and omission curves show that the performance of the model is fair. Since our model is for Oedaleus decorus asiaticus Bei-Bienko and is limited to the spatial extent of the Mongolian plateau, there is a real limitation in model extrapolation, and only a small range of model extrapolation can be performed.

  • Line 422. Why were GPS records from Mongolia not used in the current study? As previously mentioned, the use of GPS points that characterise the niche of the focal taxon most fully, is desirable and will alter the results and predictions obtained from the fitted models. Without data from other regions where the grasshopper is known to occur, the authors need to acknowledge the limitations of the current dataset if they are going to make predictions into novel regions.

[Response] Due to the inter-country policy, the grasshopper points in Mongolia are difficult to obtain, and there are few grasshopper studies on the internet about Mongolia, so we do not have access to grasshopper points in Mongolia at present. In future studies, we will try to obtain grasshopper occurrences in Mongolia. In addition, we also have the additional goal of whether we can predict the occurrence of grasshoppers in Mongolia from grasshopper occurrence sites in Inner Mongolia. We will acknowledge the shortcomings of the current data in the discussion section.

  • Line 425-426. They can be used, but it doesn't mean that they should. Why not use the GPS points from Mongolia to perform a validation of the performance of the MaxEnt model fitted by the authors? If model performance is high when projected and validated on Mongolian data, this would provide good evidence for the transferability and usability of the current model.

[Response] Thank you for your valuable suggestions. We have added omission curves and TSS. The results of AUC, TSS and omission curves show that the performance of the model is fair. Due to the inter-country policy, the grasshopper points in Mongolia are difficult to obtain, and there are few grasshopper studies on the internet about Mongolia, so we do not have access to grasshopper points in Mongolia at present. We will acknowledge the shortcomings of the current data in the discussion section.

Once again we thank you for your all comments and suggests and we hope you will find our revision satisfactory for publication in Insects.

Reviewer 4 Report

Reviewer’s comments:

The paper is based on only grasshopper occurrence data from Inner Mongolia (China) and extrapolated to whole Mongolian Plateau (including Mongolia). Corrections need in the title and text to clarify area of investigation. Please, remember that Oedaleus decorus asiaticus is also abundant in Tuva Republic of Russia (see: Sergeev et al, 2020, Far Eastern Entomologist, 402: 1-36. https://doi.org/10.25221/fee.402.1).      

I believe that the title of paper, namely, “Analysis of Spatiotemporal Variation in Habitat Suitability for Oedaleus decorus asiaticus on the Mongolian Plateau Using Maxent and Multi-source Remote Sensing Data” correctly may be changed as follow:

“Analysis of Spatiotemporal Variation in Habitat Suitability for the Grasshopper Oedaleus decorus asiaticus on the Chinese Part of Mongolian Plateau Using Maxent and Multi-source Remote Sensing Data” in order to specially stress the group of insects (grasshoppers) and studied area.

Please, include in text (at least in Discussion) and in References list the follow just published in Insects (2023) paper on habitat of Oedaleus decorus asiaticus:

Zhongxiang Sun, Huichun Ye, Wenjiang Huang, Erden Qimuge, Huiqing Bai, Chaojia Nie, Longhui Lu, Binxiang Qian and Bo Wu. (2023) Assessment on Potential Suitable Habitats of the Grasshopper Oedaleus decorus asiaticus in North China based on MaxEnt Modeling and Remote Sensing Data. Insects, 14, 138. https://doi.org/10.3390/insects14020138

Author Response

Response to Reviewer 4's comments

Thank you very much for your comments and suggestions. Your useful feedback has helped us to improve our paper substantially. We hope our responses address your concerns.

  • The paper is based on only grasshopper occurrence data from Inner Mongolia (China) and extrapolated to whole Mongolian Plateau (including Mongolia). Corrections need in the title and text to clarify area of investigation. Please, remember that Oedaleus decorus asiaticus is also abundant in Tuva Republic of Russia (see: Sergeev et al, 2020, Far Eastern Entomologist, 402: 1-36. https://doi.org/10.25221/fee.402.1).

[Response] Thank you for your comments. We restricted the scope of the Mongolian Plateau in the 2.1 (Study Area) and defined the Mongolian Plateau as Inner Mongolia of China and Mongolia. We have reviseded the title to" Analysis of Spatiotemporal Variation in Habitat Suitability for Oedaleus decorus asiaticus Bei-Bienko on the Mongolian Plateau Using Maxent and Multi-source Remote Sensing Data."

  • Please, include in text (at least in Discussion) and in References list the follow just published in Insects (2023) paper on habitat of Oedaleus decorus asiaticus: Zhongxiang SUN, Huichun YE, Wenjiang HUANG, Erden QIMUGE, Huiqing BAI, Chaojia NIE, Longhui LU, Binxiang QIAN and Bo WU. (2023) Assessment on Potential Suitable Habitats of the Grasshopper Oedaleus decorus asiaticus in North China based on MaxEnt Modeling and Remote Sensing Data. Insects, 14, 138. https://doi.org/10.3390/insects14020138

[Response] We have added this reference inside the article.

Once again, we thank you for your all comments and suggests and we hope you will find our revision satisfactory for publication in Insects.

Reviewer 5 Report

Review for ‘Analysis of spatiotemporal variation in habitat suitability for Oedaleus decorus asiaticus…’

The authors use MaxEnt to investigate whether environmental and biotic features that the authors identify are associated with presence of the pest grasshopper O. decorus. The area of concern is not easy to survey, and so the authors use remote sensing data and web archives for many of their quantitative environmental and quantitative or qualitative biotic features. However there are assumptions in using MaxEnt which are not presented, including that the insect populations are at equilibrium, which is not likely to be valid for an outbreak species such as O. decorus. Mean annual land surface temperature is likely to be highly correlated with elevation, and vegetation cover is likely to be highly correlated with biomass, such that they could present a problem for parameter selection by the models. Some environmental parameters, such as mean annual precipitation and mean annual land surface temperature, are constructed too generally to be interpreted as constraints on life-history traits as they are in the Discussion. Specific comments follow:

l. 16 change ‘analyses’ to ‘analyze’

l. 26 This is not a sentence

l. 41-42 This sentence repeats the previous sentence.

l. 44 km2 should be superscript 2

l. 48 change to ‘head’

l. 52 Explain how grasshoppers cause grassland degradation

l. 57-85 In these two paragraphs especially, the manuscript could use editing.

l. 101 change ‘was’ to ‘were’

l. 144 How to deal with biased sampling in MaxEnt? See Philips et al. 2009 Ecological Applications for an example of target-group occurrence data as background points in MaxEnt to reduce bias in presence data.

l. 161 Indicate here whether you are using annual means.

Table 1 Are land surface temperatures highly correlated with altitude? If correlations exceed 0.7, collinearity can be a problem for modeling. The year 2020 should only be listed once. It occurs in the range 2017-2021 and repeated for the decadal years: 2000, 2010, 2020.

 l. 168 ‘According’ is not the correct word to start this sentence.

l. 177 The definition for VC is confusing. It is unclear whether this is being calculated in each grid cell for May-August, and if so, is NDVI the value for that time period in that cell, and NDVI the value across all time periods in that cell.

l. 178 Are AGB and VC highly correlated. Suggest indicating any problems with collinearity as suggested in Table 1 above.  

l. 192 Spell out “European Space Agency’ if that is what ESA refers to.

l. 194 Please clarify how the Globe Cover 2009 data set was used to mask all the data. What does this mean and what is the rationale/justification?

l. 203 The original paper should be cited for the formulae rather than a secondary reference.

l. 262 Please clarify how the simulation results were excellent.

l. 273 How are these evaluated? What are the criteria for stopping to add input variables? Why shouldn’t LST and vegetation cover also be excluded?

l. 293 I thought 0.5 was considered random in MaxEnt, and so a factor equal to 0.5 or less would be considered ‘unsuitable’ rather than ‘low suitability’. Then the next level would be low suitability, and the one above that ‘highly suitable’

l. 294 Change ‘Arding’ to According

l. 295 These three grass types seem redundant. Only the words are shifted around. Can these types be found in Figure 3a?

Fig. 3 The graph in the upper left is a histogram and not a response curve. Please explain the histogram in the Figure legend. What are the blue and red bars?

l. 306 The fact that there are two suitability indices is very confusing. .I think you need to rename one of them. The SI is derived from the habitat suitabity index in 3.1.1, and so maybe it should be renamed. I also find it confusing that low suitability is given a ‘1’ when it is not different from random. Shouldn’t it’s value be 0, and then moderate suitability be given a 1 and high suitability a 2.

l. 335 How is significance evaluated?

l. 351 It’s unclear how these relationships are evaluated. Are you comparing mapped outputs, such as Fig. 5 a,c,e, to Fig. 4?

l. 357 There must be an error in this sentence. How can you combine LST and AP to make the main factor and then have AP alone be the main factor?

l. 364 change ; to :

l. 374-383 I would find a discussion of the results of the training data and the validation data more useful than this repeat of the Methods section.

l. 388 Wouldn't it be more informative to split LST into four quarters to account for the seasonality of the plateau?

l. 400 I did not find any statistical analyses. Were they omitted?

l. 410 I think with only one growth period, it is very difficult to draw accurate conclusions. Note models assume populations are in equilibrium (Jarnevich 2015 Ecological Informatics), which is probably not the case for an outbreak species like this one.

l. 416 Your parameter is mean annual precipitation, and so you can't draw the conclusion that precipitation limits spawning, hatching, AND nymphal growth. Like LST, precipitation would be best evaluated in quarters of which three of the quarters correspond to one of each of the following: spawning, hatching, and nymphal growth.

l. 431 This statement needs at least one cited reference.

Lines 443-452 This is a repeat of the Results and doesn't add to the Conclusion at all.

Author Response

Response to Reviewer 5's comments

Thank you very much for your comments and suggestions. Your useful feedback has helped us to improve our paper substantially. We hope our responses address your concerns.

  • Line 16. change ‘analyses’ to ‘analyze’.

[Response] We have changed " analyses " to " analyze ".

  • Line 26. This is not a sentence.

[Response] We have reviseded the sentence to " The key environmental variables affecting the distribution of grasshoppers and their contribution: grass type (51.3%), accumulated precipitation (24.9%), altitude (13.0%), vegetation coverage (6.6%) and land surface temperature (4.2%).".

  • Line 41-42. This sentence repeats the previous sentence.

[Response] We have reviseded the sentence to " In addition, outbreaks of grasshoppers can induce major harm to the environment and human society, such as grassland degradation and desertification ".

  • Line 44. km2 should be superscript 2.

[Response] We have changed " km2 " to " km2 ".

  • Line 48. change to ‘head’.

[Response] We have changed " heads " to " head ".

  • Line 52. Explain how grasshoppers cause grassland degradation.

[Response] grasshoppers can munch on grassland plants in large numbers, which can lead to grassland degradation. We have reviseded the sentence to " However, because the area of grassland in this region is vast and management is frequently not optimal, grasshoppers can munch on grassland plants in large numbers, which can lead to grassland degradation ".

  • change ‘was’ to ‘were’.

[Response] We have changed " was " to " were ".

  • Line 144. How to deal with biased sampling in MaxEnt? See Philips et al. 2009 Ecological Applications for an example of target-group occurrence data as background points in MaxEnt to reduce bias in presence data.

[Response] We used 1745 grasshopper occurrence points as background points for the maxent model, thus reducing the bias that occurs. At the same time, we took the average of 15 replications as the result, which also reduced the bias to some extent.

  • Line 161. Indicate here whether you are using annual means.

[Response] We have reviseded the sentence to " Therefore, we used the annual means of the following eight environmental variables to evaluate the area of suitable habitat for Oedaleus decorus asiaticus Bei-Bienko: altitude, slope, grass type, soil type, vegetation coverage, aboveground biomass, accumulated precipitation, and land surface temperature (Table 1). ".

  • Table 1 Are land surface temperatures highly correlated with altitude? If correlations exceed 0.7, collinearity can be a problem for modeling. The year 2020 should only be listed once. It occurs in the range 2017-2021 and repeated for the decadal years: 2000, 2010, 2020.

[Response] Thank you for your valuable comments, we reduced the appearance of 2020 once. We did Pearson correlation analysis for each environmental variable. The results show that land surface temperature and altitude are not strongly correlated. Meanwhile, the purpose of our simulation by maxent was to evaluate the weights of each habitat factor of grasshopper, and in the maxent model, the jackknife would have differentiated the contribution of the two indicators, which attenuated the effect of strong correlation on the model results.

  • Line 168. According’ is not the correct word to start this sentence.

[Response] We have reviseded the sentence to " Combining to the environmental factors affecting the growth and development of grasshoppers on the Mongolian Plateau, the remote sensing data used in this study were mean vegetation coverage from May to August, mean aboveground biomass from May to August, mean annual land surface temperature from January to December, and mean annual accumulated precipitation from January to December.".

  • Line 177. The definition for VC is confusing. It is unclear whether this is being calculated in each grid cell for May-August, and if so, is NDVI the value for that time period in that cell, and NDVI the value across all time periods in that cell.

[Response] We used the average NDVI data from May to August. The average NDVI data for each grid cell were used to calculate the average VC for that grid cell.

  • Line 178. Are AGB and VC highly correlated. Suggest indicating any problems with collinearity as suggested in Table 1 above.

[Response] Thank you for your valuable comments. We add to the correlation analysis in subsection 2.2.2. We have added the sentence as "To avoid strong collinearity between variables, we did a Pearson correlation analysis for each variable. The statistical analysis was implemented in SPSS Statistics 22.0 (IBM Corp, Armonk, NY, USA). The results of the analysis showed that vegetation coverage was strongly correlated with above ground biomass. However, the purpose of our simulation by maxent was to evaluate the weights of each habitat factor of grasshopper, and in the maxent model, the jackknife would have differentiated the contribution of the two indicators, which attenuated the effect of strong correlation on the model results. Meanwhile, since overgrazing is more common in the study area, this further reduces the correlation between the variables."

  • Line 192. Spell out “European Space Agency’ if that is what ESA refers to.

[Response] We have changed " ESA " to " European Space Agency ".

  • Line 194. Please clarify how the Globe Cover 2009 data set was used to mask all the data. What does this mean and what is the rationale/justification?

[Response] Our aim was to obtain the grasshopper inhabitable area in the grassland areas of the Mongolian plateau. Therefore, we reclassified the Globe Cover 2009 data by ArcGIS software to represent the Mongolian Plateau grassland area with 1 and the non-grassland area with 0 (white area in Fig. 4) and masked the grasshopper inhabitable area of the Mongolian Plateau grassland by ArcGIS software. The grassland areas were distinguished from non-grassland areas in the grasshopper inhabitable area of the Mongolian Plateau for each period.

  • Line 203.The original paper should be cited for the formulae rather than a secondary reference.

[Response] We have made changes to this.

  • Line 262. Please clarify how the simulation results were excellent.

[Response] We have added omission curves and TSS. The results showed an AUC value of 0.910 and a TSS value of 0.631, and the mean omission on test samples is close to the predicted omission. These results indicate that the simulation results were fair. We have reviseded the sentence to " In addition, the average TSS value for 15 randomly repeated runs was 0.631, the mean omission on test samples is close to the predicted omission (the black straight line). These results indicate that the simulation results were fair (Figure 2)".

  • Line 273. How are these evaluated? What are the criteria for stopping to add input variables? Why shouldn’t LST and vegetation cover also be excluded?

[Response] We selected variables with high contribution and importance as key environmental variables based on the assessment results of the maxent model on the contribution and importance of each environmental variable. Land surface temperature and vegetation coverage had high contribution or importance relative to soil type, slope, and above ground biomass. Therefore, we did not exclude vegetation coverage and land surface temperature in order to improve the completeness of the overall assessment index.

  • Line 293. I thought 0.5 was considered random in MaxEnt, and so a factor equal to 0.5 or less would be considered ‘unsuitable’ rather than ‘low suitability’. Then the next level would be low suitability, and the one above that ‘highly suitable’

[Response] In the maxent model,0.5indicates the probability of grasshopper occurrence; below 0.5 is not non-occurrence, and we have added references in the text to explain this.

  • Line 294. Change ‘Arding’ to According

[Response] We have changed " Arding " to " According ".

  • Line 295. These three grass types seem redundant. Only the words are shifted around. Can these types be found in Figure 3a?

[Response] These three grassland types represent three different types of grassland, which can be found in Figure 3a and which we have illustrated below Table 5. We have reviseded the sentence to "According to the classification results of the habitat factors (Table 4), habitat suitability was highest when the grass types were temperate desertified grassland, temperate steppe desert, and temperate desert".

  • 3 The graph in the upper left is a histogram and not a response curve. Please explain the histogram in the Figure legend. What are the blue and red bars?

[Response] Depending on the data type, different data formats (Continuous or Categorical) are set in maxent, where the Continuous format outputs a continuous response curve, and the Categorical format outputs a histogram. The curves show the mean response of the 15 replicate Maxent runs (red) and the mean +/- one standard deviation (blue). We provide an explanation at the bottom of Figure 3

  • Line 306. The fact that there are two suitability indices is very confusing. .I think you need to rename one of them. The SI is derived from the habitat suitabity index in 3.1.1, and so maybe it should be renamed. I also find it confusing that low suitability is given a ‘1’ when it is not different from random. Shouldn’t it’s value be 0, and then moderate suitability be given a 1 and high suitability a 2.

[Response] We have changed " suitability index (SI) " to " inhabitability index(IH) ". In our study, non-grassland areas and areas with a habitability index less than 1 were defined as unsuitable areas (assigned a value of 0), areas with (1.0 ≤ IH < 1.5) were defined as low suitability areas (assigned a value of 1), areas with (1.5 ≤ IH < 2.5) were defined as medium suitability areas (assigned a value of 2), and areas with (2.5 ≤ IH < 3) were defined as high suitability areas (assigned a value of 3).

  • Line 335. How is significance evaluated?

[Response] The assessment was based on the extent of changes in the area and distribution of the grasshopper inhabitable area in the Mongolian plateau in 2000, 2010 and 2020. From Figure 4, the change in the grasshopper inhabitable area in 2000 compared to 2010 was not significant, while 2020 showed a large change compared to 2000 and 2010. We have reviseded the sentence to " The distribution of suitable habitat for grasshoppers on the Mongolia Plateau was similar in 2000 and 2010, but the distribution of suitable habitat for grasshoppers in 2020 was significantly different from that in 2000 and 2010 (Figure 4).".

  • Line 351. It’s unclear how these relationships are evaluated. Are you comparing mapped outputs, such as Fig. 5 a,c,e, to Fig. 4?

[Response] we are comparing mapped outputs, such as Fig. 5. Figure 5 shows the distribution of the suitability of accumulated precipitation (AP) and land surface temperature (LST) for grasshoppers on the Mongolian Plateau in 2000, 2010, and 2020. By analyzing the changes in land surface temperature and accumulated precipitation on grasshopper suitability in Figure 5, the main drivers of changes in grasshopper inhabitable area can be obtained.

  • Line 357. There must be an error in this sentence. How can you combine LST and AP to make the main factor and then have AP alone be the main factor?

[Response] We have removed this sentence " Combined with the land surface temperature and accumulated precipitation, the main habitat factor driving large changes in habitat suitability for grasshoppers on the Mongolian Plateau in 2020 was accumulated precipitation."

  • Line 364. change ; to :

[Response] We have changed “; " to " : ".

  • Line 374-383. I would find a discussion of the results of the training data and the validation data more useful than this repeat of the Methods section.

[Response] We have reviseded the sentence to " Maxent models are based on Maxent theory, and they can be used to predict the potential distribution of species through occurrence records and environmental variables[50]. Maxent models simplify complex natural systems and permit reliable predictions to be obtained with small sample sizes[51,52]. The results of the maxent model simulation showed that the average AUC value for 15 repeated runs was 0.910 and the TSS value was 0.631. Meanwhile, the average omission rate of the test sample was close to the predicted omission rate as shown by the results of the omission curve. The above results indicate that the model has a good performance. However, since grasshoppers are an outbreak species, this may reduce the accuracy of the model. In addition, since our model is for Oedaleus decorus asiaticus Bei-Bienko and is limited to the spatial extent of the Mongolian plateau, there is a real limitation in model extrapolation, and only a small range of model extrapolation can be performed, although clamping is possible. If you want to extrapolate the model, you need to adjust the extrapolation area, for example,Multi-surface Environmental Similarity Surfaces (MESS) could be computed[66]."

  • Line 388. Wouldn't it be more informative to split LST into four quarters to account for the seasonality of the plateau?

[Response] Thank you for your valuable comments. It is indeed informative to consider the surface temperature in four seasons, because it is possible to determine at which stage the surface temperature has a greater impact on the grasshopper. In this study, we aimed to assess the effect of land surface temperature on the entire reproductive period of grasshoppers, therefore, did not consider land surface temperature in four seasons. In addition, the aim of this study was to extract the inhabitable area of grasshoppers in the steppe of Mongolian plateau, which is a more macroscopic study. In the next study, we will study the four growth stages of the grasshopper, and we will consider the effect of land surface temperature on the growth stages of the grasshopper. Thank you again for your valuable comments.

  • Line 400. I did not find any statistical analyses. Were they omitted?

[Response] We performed statistical analysis in subsections 3.2(3.2 Habitat Suitability for Grasshoppers on the Mongolian Plateau) and 3.3(3.3. Spatial and Temporal Variability in Habitat Suitability for Grasshoppers on the Mongolian Plateau and the Effects of Environmental Factors). The results show that no significant difference in the distribution of areas with suitable habitats for grasshoppers was observed between 2000 and 2010.

  • Line 410. I think with only one growth period, it is very difficult to draw accurate conclusions. Note models assume populations are in equilibrium (Jarnevich 2015 Ecological Informatics), which is probably not the case for an outbreak species like this one.

[Response] Thank you for your valuable comments. We arrived at this conclusion by comparing the results of the calculation of the inhabitable area of grasshoppers in the Mongolian plateau in 2000, 2010 and 2020 (Figure 4). The results of the calculation of the grassland locust habitat on the Mongolian plateau show that the area of high suitability for grassland locusts in 2020 is more than that in 2000 and 2010. Second, the maxent model assumes that the species is in equilibrium, while grasshoppers are an outbreak species, which may indeed lead to possible bias in the simulation results. However, there have been studies using the maximum entropy model to simulate the distribution of grasshoppers, and their simulation results are more reasonable. Meanwhile, we used 5 years of grasshopper occurrence points as the background points of the model, which can basically represent the areas with more stable locust distribution over this year.

  • Line 416. Your parameter is mean annual precipitation, and so you can't draw the conclusion that precipitation limits spawning, hatching, AND nymphal growth. Like LST, precipitation would be best evaluated in quarters of which three of the quarters correspond to one of each of the following: spawning, hatching, and nymphal growth.

[Response] Thank you for your valuable comments. It is indeed informative to consider the accumulated precipitation in four seasons, because it is possible to determine at which stage the accumulated precipitation has a greater impact on the grasshopper. In this study, we aimed to assess the effect of accumulated precipitation on the entire reproductive period of grasshoppers, therefore, did not consider accumulated precipitation in four seasons. In addition, the aim of this study was to extract the inhabitable area of grasshoppers in the steppe of Mongolian plateau, which is a more macroscopic study. In the next study, we will study the four growth stages of the grasshopper, and we will consider the effect of accumulated precipitation on the growth stages of the grasshopper. Thank you again for your valuable comments. We have removed the Sentence " Previous studies have shown that precipitation affects the occurrence of grasshoppers by affecting soil moisture [29], and the effects of precipitation on the growth of grasshoppers vary among growth stages. In the spawning stage, hatching occurs, and grasshopper nymphs are unearthed; precipitation limits the distribution of grasshoppers."

  • Line 431. This statement needs at least one cited reference.

[Response] We have added two references.

  • Line 443-452. This is a repeat of the Results and doesn't add to the Conclusion at all.

[Response] Thank you for your valuable comments. We have reviseded the sentence to " In this study, eight habitat factors: altitude, slope, grass type, soil type, vegetation coverage, aboveground biomass, accumulated precipitation, and land surface temperature, were used to simulate the distribution of suitable habitat for the dominant grasshopper species on the Mongolian Plateau from 2017 to 2021 using Maxent and remote sensing data. Five key habitat factors (grassland type, accumulated precipitation, altitude, vegetation coverage, land surface temperature) with the strongest effects on the habitat suitability for grasshoppers on the Mongolian Plateau were identified, and they were used to characterize long-term temporal and spatial changes in habitat suitability for grasshoppers on the Mongolian Plateau in 2000, 2010, and 2020. The results showed that the area of grasshopper high suitability areas is larger in 2020 than in 2000 and 2010, mainly in the southeastern, central and northern parts of the Mongolian Plateau; the distribution of areas with suitable habitat for grasshoppers was similar in 2000 and 2010;the main drivers of changes in grasshopper inhabitable area in the Mongolian Plateau are accumulated precipitation and land surface temperature. The results of this study will aid the monitoring of grasshopper populations, including the vulnerability of different regions to grasshopper outbreaks, on the Mongolian Plateau in the future.".

Once again we thank you for your all comments and suggests and we hope you will find our revision satisfactory for publication in Insects.

Round 2

Reviewer 1 Report

It is only needed small improvements in the text and figure description.

The recommendations are in the attached file

Author Response

Thank you very much for your comments and suggestions. Your useful feedback has helped us to improve our paper substantially. We hope our responses address your concerns.

  • Line 18. It is only needed to mention the complete name with taxonomic classifier the first time that appears in the text, next times could be as follows O. decorus asiaticus.

[Response] Thank you for your valuable comments. We have changed " Oedaleus decorus asiaticus Bei-Bienko " to " O. decorus asiaticus " in the article.

  • Line 25. To make more clear the information I recommend to add numbers or letters to identify every map, for example A) Locations of ... B)Altitude in the..... C)Grass type...Also add the letter in the corner or below every figure.

[Response] Thank you for your valuable comments. We have introduced the images in the legend of each image; therefore, we have not numbered the images.

Once again we thank you for your all comments and suggests and we hope you will find our revision satisfactory for publication in Insects.

Reviewer 2 Report

The manuscript is sufficiently improved. One minor thing, I suggest using abbreviating the species name after it is mentioned in full at fist appearance.

Best wishes,

Author Response

Thank you very much for your comments and suggestions. Your useful feedback has helped us to improve our paper substantially. We hope our responses address your concerns.

  • The manuscript is sufficiently improved. One minor thing, I suggest using abbreviating the species name after it is mentioned in full at fist appearance.

[Response] Thank you for your valuable comments. We have changed " Oedaleus decorus asiaticus Bei-Bienko " to " O. decorus asiaticus " in the article.

Once again we thank you for your all comments and suggests and we hope you will find our revision satisfactory for publication in Insects.

Reviewer 5 Report

Capitalize Maxent throughout.

l. 27 This is still not a sentence. There is no verb. Add ‘were’ after ‘contribution’

l. 53 ‘because the area of grassland in this region is vast…, grasshoppers can munch on grassland plants in large numbers, which can lead to grassland degradation.’ It seems the management is leading to ‘grassland degradation’ since it is frequently not optimal. The fact the area is vast should not affect grasshopper feeding, but grasshopper densities would. Maybe rewrite as ‘However, management is frequently not optimal in this vast area of grassland, and grasshopper  populations can be sufficiently dense to result in additional habitat degradation.’

L.56 ‘invert’ is not the correct word. Do you mean ‘infer’?

l. 63 italicize Leymus chinensis

l. 142 The grass types are in the key on the map and so do not need to be repeated. Instead, I would indicate that there is a map of grassland and desert types. A definition of background value would be useful in the legend.

l. 148 Can the ‘rules’ be described in brief here? If this is a brief description, then just start with 'In brief,' before ‘a total of 1745…’. Are occurrence points GPS points? Weren't the numbers and locations of grasshoppers recorded during the surveys? What happens to the GPS points where 'zero' grasshoppers were surveyed? In other words, why are only occurrence records kept?

L. 174 Where is the result for the correlation analysis? You need to include a Table of the correlations between the factors. I believe this explanation of jackknifing differentiating the contribution of the two collinear variables is weak. It definitely needs a citation or two to strengthen the statement.

l. 179 I don't understand how overgrazing would reduce the correlation between vegetation cover and biomass. It seems like it would enhance the correlation. Anyway it too is a weak explanation to justify highly collinear variables. Selection of one by Maxent is very likely to prevent selection of the other and yet both are probably of equal importance. 

l. 215 You still haven’t explained ‘Globe cover dataset was used to mask all the data’ in the paper like you did in the Reply to Reviewer.

l. 352 Fig. 3 legend So the reader does not have to refer back to Table 1 or 3, define GT, AP, VC and LST in the figure legend.

l. 379 superscript ‘2’

l. 393 Where is the test showing that they are 'significantly different'? You indicate that there is a test, but I think you are comparing plots, which are just ‘qualitative differences’ and not ‘statistical differences’

l. 403 Same question: Where is the test for significant differences? Are these just 'qualitative differences'? Or is there a quantitative statistical test somewhere?

l. 436 Indicate that the model assumes the population is in equilibrium.

l. 440 Define ‘clamping’.

l. 441 The sentences ‘If you want to extrapolate the model, you need to adjust the extrapolation area, for exampleMulti-surface Environmental Similarity Surfaces (MESS) could be computed [64].’ This is advice that the authors should have followed. Please revise the sentence to omit the ‘you’ subjects. Also can the authors verify that they extrapolated the model within the ‘small range’ permitted by Maxent.

l. 480 ‘survey points may deviate from the GPS data’ Do you mean that the survey point may not have an accurate GPS location? Could you state it that way? I expect this is a small source of error since most of the points are in one region of a vast area that you are extrapolating to.

l. 507 Do reviewers have access to the following: Figure S1: title; Table S1: title; Video S1: title. I did not see any supplementary material to review.

Author Response

Thank you very much for your comments and suggestions. Your useful feedback has helped us to improve our paper substantially. We hope our responses address your concerns.

  • Line 27. This is still not a sentence. There is no verb. Add ‘were’ after ‘contribution’.

[Response] Thank you for your valuable comments. We have added " were " after ‘contribution’.

  • Line 53. because the area of grassland in this region is vast…, grasshoppers can munch on grassland plants in large numbers, which can lead to grassland degradation.’ It seems the management is leading to ‘grassland degradation’ since it is frequently not optimal. The fact the area is vast should not affect grasshopper feeding, but grasshopper densities would. Maybe rewrite as ‘However, management is frequently not optimal in this vast area of grassland, and grasshopper populations can be sufficiently dense to result in additional habitat degradation.’

[Response] Thank you for your valuable comments. We have reviseded the sentence to " However, management is frequently not optimal in this vast area of grassland, and grasshopper populations can be sufficiently dense to result in additional habitat degradation.".

  • Line 56. invert’ is not the correct word. Do you mean ‘infer’?

[Response] We have changed " invert " to " infer " in the sentence.

  • Line 63. italicize Leymus chinensis.

[Response] We have changed " Leymus chinensis " to " Leymus chinensis ".

  • Line 142. The grass types are in the key on the map and so do not need to be repeated. Instead, I would indicate that there is a map of grassland and desert types. A definition of background value would be useful in the legend.

[Response] Thank you for your valuable comments. We have removed the repetitive content. In addition, the legend for grass types shows the different types of grass. We have only researched grasslands in the Mongolian Plateau, so we do not have information on deserts in the Mongolian Plateau region.

  • Line 148. Can the ‘rules’ be described in brief here? If this is a brief description, then just start with 'In brief,' before ‘a total of 1745…’. Are occurrence points GPS points? Weren't the numbers and locations of grasshoppers recorded during the surveys? What happens to the GPS points where 'zero' grasshoppers were surveyed? In other words, why are only occurrence records kept?

[Response] Thank you very much for your valuable comments. Our approach to collecting grasshopper points is based on the agricultural industry standards of the People's Republic of China. We have reviseded the sentence to " In brief, a total of 1,745 grasshopper occurrence points were obtained by route survey method for spring, summer and autumn seasons from 2018 to 2022, and the distance be-tween sampling points was not less than 100m. ". The grasshopper points we obtained were GPS locations, and the number and location of grasshoppers at the point were recorded during the survey. In addition, our approach is to invert the grasshopper habitat zone on the Mongolian Plateau using key grasshopper habitat factors, so we only need to obtain points where grasshoppers occur and use a maximum entropy model to construct relationships between grasshopper points and habitat factors to invert the grasshopper habitat zone on the Mongolian Plateau.

  • Where is the result for the correlation analysis? You need to include a Table of the correlations between the factors. I believe this explanation of jackknifing differentiating the contribution of the two collinear variables is weak. It definitely needs a citation or two to strengthen the statement.

[Response] Thank you for your valuable comments. We added a correlation analysis table (Table 2), and the results show a strong correlation between above ground biomass, vegetation coverage and land surface temperature. As the aim of our study is to assess the weight of each habitat factor on grasshopper occurrence, to obtain the key habitat factors for grasshopper occurrence, and then to use the key habitat factors to invert the suitable areas for grasshoppers on the Mongolian Plateau, this approach reduces the influence of strong correlations on the study results. In addition, the jackknife method further reduces the influence of strong correlations by separating each habitat factor independently when assessing the weight of each habitat factor. At the same time, we removed some habitat factors with low weights when using them to invert the Mongolian Plateau grasshopper habitat zone from which above ground biomass was removed. In addition, we did not remove land surface temperature due to its high importance (Table 4). This may make our study somewhat flawed, and we will refine our research approach more in later studies.

  • Line 179. I don't understand how overgrazing would reduce the correlation between vegetation cover and biomass. It seems like it would enhance the correlation. Anyway it too is a weak explanation to justify highly collinear variables. Selection of one by Maxent is very likely to prevent selection of the other and yet both are probably of equal importance.

[Response] The point we are trying to make is that overgrazing leads to a reduction in vegetation coverage and above ground biomass, which in turn leads to a reduction in the effect of vegetation coverage and above ground biomass on grasshopper occurrence. Therefore, the effect of the strong correlation between above ground biomass and vegetation coverage will be reduced. The jackknife will assess the importance of each habitat factor and the results show that strongly correlated variables do not have the same importance as each other (Table 4).

  • Line 215. You still haven’t explained ‘Globe cover dataset was used to mask all the data’ in the paper like you did in the Reply to Reviewer.

[Response] We have reviseded the sentence to " To distinguish the steppe areas of the Mongolian plateau from the non-steppe areas, the Globe Cover 2009 dataset with 0 and 1 attributes was used to mask all the data (0 for non-grassland areas, 1 for grassland areas).".

  • Line 352. Fig. 3 legend So the reader does not have to refer back to Table 1 or 3, define GT, AP, VC and LST in the figure legend.

[Response] We have reviseded the sentence to " GT for grass type, AP for accumulated precipitation, VC for vegetation coverage and LST for land surface temperature. ".

  • Line 379. superscript ‘2’

[Response] We have changed "km2" to "km²."

  • Line 393. Where is the test showing that they are 'significantly different'? You indicate that there is a test, but I think you are comparing plots, which are just ‘qualitative differences’ and not ‘statistical differences’

[Response] Thank you for your valuable comments. Our aim is to obtain the main drivers of change in the distribution of grasshopper habitats on the Mongolian Plateau, so it is more intuitive to compare the differences in the images. Also, in section 3.2, we calculate the area of the grasshopper habitat in the Mongolian Plateau grassland in 2000, 2010 and 2020, from which we can see that the distribution of the grasshopper habitat in the Mongolian Plateau grassland in 2020 differs significantly from that in 2000 and 2010.

  • Line 403. Same question: Where is the test for significant differences? Are these just 'qualitative differences'? Or is there a quantitative statistical test somewhere?

[Response] Thank you for your valuable comments. The aim here is to identify the main drivers of change in the distribution of the grassland locust niche on the Mongolian Plateau, which can be seen visually in Figures 4 and 5. These are 'qualitative differences' and it is difficult to use quantitative statistics to explain changes in the distribution of grassland locust habitats across the Mongolian Plateau. Therefore, we have not used quantitative statistics here.

  • Line 436. Indicate that the model assumes the population is in equilibrium.

[Response] We have reviseded the sentence to " However, since grasshoppers are an outbreak species, the Maxent model assumes that species are in equilibrium, this may reduce the accuracy of the model. ".

  • Line 440. Define ‘clamping’.

[Response] Inappropriate use of words. We have removed the word.

  • Line 441. The sentences ‘If you want to extrapolate the model, you need to adjust the extrapolation area, for exampleMulti-surface Environmental Similarity Surfaces (MESS) could be computed [64].’ This is advice that the authors should have followed. Please revise the sentence to omit the ‘you’ subjects. Also can the authors verify that they extrapolated the model within the ‘small range’ permitted by Maxent.

[Response] We have reviseded the sentence to " If want to extrapolate the model, need to adjust the extrapolation area, for example,Multi-surface Environmental Similarity Surfaces (MESS) could be computed ". At present, due to time issues, we have not yet confirmed that they extrapolated the model within the ‘small range’ permitted by Maxent. So, we removed the sentence "and only a small range of model extrapolation can be performed, although clamping is possible". We will confirm this in a later study.

  • Line 480. survey points may deviate from the GPS data’ Do you mean that the survey point may not have an accurate GPS location? Could you state it that way? I expect this is a small source of error since most of the points are in one region of a vast area that you are extrapolating to.

[Response] Our grasshopper locations are accurate GPS locations. What we mean by this is that there may be some small errors in grasshopper locations due to operator error.

  • Line 507. Do reviewers have access to the following: Figure S1: title; Table S1: title; Video S1: title. I did not see any supplementary material to review.

[Response] This content may require you to communicate with the editor.

Once again we thank you for your all comments and suggests and we hope you will find our revision satisfactory for publication in Insects.

Round 3

Reviewer 5 Report

I really appreciate what you have done, but you need to talk to a statistician. A correlation of 0.995 between two variables is indicating that the two variables are interchangeable. It doesn’t matter what jackknife method you employ, VC and AGB are measuring the same thing.  Highly correlated parameters like LST with AGB (0.90), LST and VC (0.911) also create problems in model selection.

Other comments

l. 148 The number and location of grasshoppers were recorded after the surveys. Don’t you mean during the surveys? Please introduce a new sentence after this one that states: ‘Absence data were not recorded.’

l. 392 The authors continue to use the word ‘significantly’ without a statistical analysis. Please change the wording to ‘qualitatively different upon visual inspection of the mapped suitability levels’

l. 402 The authors indicate a lack of significance without a statistical test. Please change the wording to ‘qualitatively similar’

Author Response

Response to Reviewer 5's comments

Thank you very much for your comments and suggestions. Your useful feedback has helped us to improve our paper substantially. We hope our responses address your concerns.

  • I really appreciate what you have done, but you need to talk to a statistician. A correlation of 0.995 between two variables is indicating that the two variables are interchangeable. It doesn’t matter what jackknife method you employ, VC and AGB are measuring the same thing. Highly correlated parameters like LST with AGB (0.90), LST and VC (0.911) also create problems in model selection.

[Response] Thank you for your valuable comments, we think the suggestions you made are reasonable. In this study, we chose vegetation coverage and land surface temperature to assess grasshopper habitability in the Mongolian Plateau for two reasons:(1) The contribution of vegetation coverage and land surface temperature were similar in the Maxent analysis results, and the importance of land surface temperature was much higher than that of vegetation coverage, indicating that although there was a strong correlation between vegetation coverage and land surface temperature, the Maxent model differentiated the contribution of the two indicators through jackknife, which attenuated the influence of the strong correlation on the model results. (2) The purpose of our Maxent simulation is to obtain the weights of each grasshopper habitat indicator in assessing grasshopper habitat suitability, not to directly use all habitat indicators to assess the suitability of grasshoppers on the Mongolian Plateau, and to filter out the indicators with high contribution and importance as key indicators (we excluded above ground biomass here) based on the Maxent model's assessment of each indicator, and use the key habitat indicators assess the suitability of Mongolian plateau grasshoppers. At the same time, the Maxent model can differentiate the contribution of vegetation coverage and land surface temperature, and the assessment of grasshopper habitat suitability by considering both vegetation coverage and land surface temperature can improve the completeness of the overall assessment index (95.5% contribution), while excluding either vegetation coverage or land surface temperature will reduce the completeness of the model assessment index. In addition, when assessing grasshopper habitability on the Mongolian Plateau, we need to consider the environmental impact on all aspects of grasshoppers (this is described in section 2.2.2). Therefore, we believe that the assessment index system we constructed can be used to assess grasshopper suitability.

  • Line 148. The number and location of grasshoppers were recorded after the surveys. Don’t you mean during the surveys? Please introduce a new sentence after this one that states: ‘Absence data were not recorded.’

[Response] Thank you for your valuable comments. Yes, we mean during the surveys. We have reviseded the sentence to " The number and location of grasshoppers were recorded after the surveys, absence data were not recorded.".

  • Line 392. The authors continue to use the word ‘significantly’ without a statistical analysis. Please change the wording to ‘qualitatively different upon visual inspection of the mapped suitability levels’

[Response] Thank you for your valuable comments. We have reviseded the sentence to " Upon visual inspection, the distribution of suitable habitat for grasshoppers on the Mongolia Plateau was similar in 2000 and 2010, but the distribution of suitable habitat for grasshoppers in 2020 was qualitatively different of the mapped suitability levels from that in 2000 and 2010 (Figure 4).".

  • Line 402. The authors indicate a lack of significance without a statistical test. Please change the wording to ‘qualitatively similar’

[Response] Thank you for your valuable comments. We have reviseded the sentence to " According to the classification of habitat factors in 2000, 2010, and 2020 (Figure 5), there were qualitatively similar in average vegetation coverage of the same grass types and at the same altitudes. ".

Once again we thank you for your all comments and suggests and we hope you will find our revision satisfactory for publication in Insects.

Round 4

Reviewer 5 Report

Allow me to repeat what I said in my last review: I really appreciate what you have done, but you need to talk to a statistician. A correlation of 0.995 between two variables is indicating that the two variables are interchangeable. It doesn’t matter what jackknife method you employ, VC and AGB are measuring the same thing.  Highly correlated parameters like LST with AGB (0.90), LST and VC (0.911) also create problems in model selection.

See Dormann et al. 2013 Collinearity: a review of methods to deal with it and a simulation study evaluating their performance. Ecography 36:27-46.

See also Section 3 in Jarnevich et al. 2015 Caveats for correlative species distribution modeling. Ecological Informatics 29:6-15.

I found this recommendation in a blog on MaxEnt:

If two variables are highly correlated (greater than 75%) I would keep the one that has the higher gain in the jackknife test. It is always interesting and useful to switch the two variables (drop the higher gain one and keep the other) and see if your outcome is similar, however. If it is not, then you may wonder about the robustness of the predicted distribution.

Other comments

l. 148 If it correct that grasshopper observations were recorded during the surveys, then change the sentence “The number and location of grasshoppers were recorded after the surveys.’ to ‘The number and location of grasshoppers were recorded during the surveys.’ Records made during the surveys would have less error than recording of data after the surveys.

l. 167 Reword the first sentence:

Change ‘To avoid strong collinearity between variables,’ to ‘To test for strong collinearity between variables..

l. 171 Delete the following:

and in the maxent model, the jackknife would have differentiated the contribution of the two indicators, which attenuated the effect of strong correlation on model results.

l. 174 Delete the sentence ‘Meanwhile, since overgrazing is more common in the study area, this further reduces the correlation between the variables [57]’. After reading [57], it is evident that the authors of that paper also failed to account for collinearity of their independent variables and introduced several explanations in the Discussion that have no data to back them up. Unfortunately, citing text without data to support it is not sufficient to justify using multiple highly correlated environmental variables in model selection.

Author Response

Response to Reviewer 5's comments

Thank you very much for your comments and suggestions. Your useful feedback has helped us to improve our paper substantially. We hope our responses address your concerns.

  • Allow me to repeat what I said in my last review: I really appreciate what you have done, but you need to talk to a statistician. A correlation of 0.995 between two variables is indicating that the two variables are interchangeable. It doesn’t matter what jackknife method you employ, VC and AGB are measuring the same thing. Highly correlated parameters like LST with AGB (0.90), LST and VC (0.911) also create problems in model selection.

[Response] Thank you for your valuable comments, we think the suggestions you made are reasonable. In this study, there was a strong correlation between above ground biomass, vegetation coverage and land surface temperature. Based on the results of the Maxent model runs, the importance and contribution of above ground biomass was low, so we removed above ground biomass from the calculations of the grasshopper’s suitable habitat on the Mongolian Plateau. However, we have retained vegetation coverage and land surface temperature due to their high importance and contribution. Based on your suggestion, we have recalculated the grasshopper habitat on the Mongolian Plateau and the results show that there is no significant difference in the results of the grasshopper habitat calculations, whether vegetation coverage or land surface temperature is removed, as the vegetation coverage and land surface temperature are given less weight compared to grass type, altitude and accumulated precipitation and do not affect the overall results much. In contrast, as grasshopper growth and development are influenced by a variety of environmental factors, we believe that considering both vegetation coverage and land surface temperature would better reflect the distribution of grasshoppers in the grasslands of the Mongolian plateau and would also improve the completeness of the overall assessment indicators (95.5% contribution). In addition, land surface temperature mainly affects grasshopper egg laying and overwintering, while vegetation coverage mainly affects grasshopper habitat selection. The two variables have different effects on grasshoppers, and it is more reasonable to consider both vegetation coverage and land surface temperature to assess the distribution of grasshopper habitats on the Mongolian Plateau. Therefore, we believe that the assessment index system we have constructed can be used to assess grasshopper habitat suitability in the Mongolian Plateau.

  • Line 148. If it correct that grasshopper observations were recorded during the surveys, then change the sentence “The number and location of grasshoppers were recorded after the surveys.’ to ‘The number and location of grasshoppers were recorded during the surveys.’ Records made during the surveys would have less error than recording of data after the surveys.

[Response] Thank you for your valuable comments. We have reviseded the sentence to " The number and location of grasshoppers were recorded during the surveys.".

  • Line 167. Reword the first sentence:Change ‘To avoid strong collinearity between variables,’ to ‘To test for strong collinearity between variables..

[Response] Thank you for your valuable comments. We have reviseded the sentence to " To test for strong collinearity between variables".

  • Line 171. Delete the following:and in the maxent model, the jackknife would have differentiated the contribution of the two indicators, which attenuated the effect of strong correlation on model results.

[Response] Thank you for your valuable comments. We have removed the following sentence: and in the maxent model, the jackknife would have differentiated the contribution of the two indicators, which attenuated the effect of strong correlation on model results.

  • Line 174. Delete the sentence ‘Meanwhile, since overgrazing is more common in the study area, this further reduces the correlation between the variables [57]’. After reading [57], it is evident that the authors of that paper also failed to account for collinearity of their independent variables and introduced several explanations in the Discussion that have no data to back them up. Unfortunately, citing text without data to support it is not sufficient to justify using multiple highly correlated environmental variables in model selection

[Response] Thank you for your valuable comments. We have removed the following sentence: Meanwhile, since overgrazing is more common in the study area, this further reduces the correlation between the variables [57].

Once again we thank you for your all comments and suggests and we hope you will find our revision satisfactory for publication in Insects.
